# Construction of emissive ruthenium(II) metallacycle over 1000 nm wavelength for in vivo biomedical applications

Yuling Xu[1,6], Chonglu Li[1,6], Shuai Lu[2,3], Zhizheng Wang[1,4], Shuang Liu[5], Xiujun Yu[2,3], Xiaopeng Li [2,3✉] & Yao Sun[1✉]

Although Ru(II)-based agents are expected to be promising candidates for substituting Pt-drug, their in vivo biomedical applications are still limited by the short excitation/emission wavelengths and unsatisfactory therapeutic efficiency. Herein, we rationally design a Ru(II) metallacycle with excitation at 808 nm and emission over 1000 nm, namely **Ru1085**, which holds deep optical penetration (up to 6 mm) and enhanced chemo-phototherapy activity. In vitro studies indicate that **Ru1085** exhibits prominent cell uptake and desirable anticancer capability against various cancer cell lines, especially for cisplatin-resistant A549 cells. Further studies reveal **Ru1085** induces mitochondria-mediated apoptosis along with S and G2/M phase cell cycle arrest. Finally, **Ru1085** shows precise NIR-II fluorescence imaging guided and long-term monitored chemo-phototherapy against A549 tumor with minimal side effects. We envision that the design of long-wavelength emissive metallacycle will offer emerging opportunities of metal-based agents for in vivo biomedical applications.

[1] Key Laboratory of Pesticides and Chemical Biology, Ministry of Education, International Joint Research Center for Intelligent Biosensor Technology and Health, College of Chemistry, Central China Normal University, Wuhan, Hubei 430079, China. [2] Shenzhen University General Hospital, Shenzhen University Clinical Medical Academy, Shenzhen, Guangdong 518055, China. [3] College of Chemistry and Environmental Engineering, Shenzhen University, Shenzhen, Guangdong 518060, China. [4] Guangdong Provincial Key Laboratory of Luminescence from Molecular Aggregates, South China University of Technology, Guangzhou, Guangdong 510640, China. [5] School of Materials Science and Engineering, Wuhan University of Technology, Wuhan, Hubei 430070, China. [6] These authors contributed equally: Yuling Xu, Chonglu Li. ✉email: xiaopengli@szu.edu.cn; sunyaogbasp@mail.ccnu.edu.cn

Ruthenium (Ru) complexes were identified as ideal chemotherapy agents for cancer treatment due to their lower nonspecific toxicities and higher activities against Pt-drug-resistant cancer cells[1–4]. More recently, the utilization of Ru(II)-polypridyl complexes as photodynamic therapy (PDT) agents has further ameliorated therapeutic efficiency to benefit anticancer practice[5–9]. Despite the achieved success, some major issues of the current Ru(II) complexes still limited their widespread in vivo applications[10,11]. First, the short excitation/emission wavelengths ($\lambda_{ex} < 600$ nm, $\lambda_{em} < 700$ nm) of reported Ru(II) complexes failed to efficiently penetrate deep tissues, which not only reduces the efficiency of phototherapy but also undermines in vivo precisely delineating lesion margin and real-time monitoring/evaluation of therapeutic effects[12,13]. Second, single chemo- or phototherapy modality of Ru(II) complexes cannot simultaneously balance efficiency and safety[14,15]. Besides, recent studies have revealed that macromolecular drugs (i.e., metallomacrocycles) possess superior cellular uptake and longer retention time in cancer cells compared to their small counterparts[16–18]. As such, developing long-wavelength emissive Ru(II) macromolecules with combinational chemo-phototherapy is highly demanded to enable accurate in vivo cancer diagnosis and therapy.

Recently, a wide range of metallomacrocycles with variable shapes, sizes, and functionalities have been constructed by leveraging the coordination-driven self-assembly approach and drawn increasing attention in biomedicine[19–23]. Among them, Ru(II) metallacycles have been successfully applied for the chemotherapeutic or phototherapeutic treatment of cancer[24–28]. Noteworthy, Ru(II) metallacycles can selectively enter cancer cells with a long-term stay compared to normal cells[29,30]. More importantly, the emission/excitation wavelength and phototherapy efficiency of metallacycles could be finely regulated by embedding well-designed fluorescent ligands without tedious chemical synthesis[31–34]. However, it still remains challenging for in vivo noninvasively monitoring Ru(II) metallacycles delivery/biodistribution and therapeutic feedback in a timely manner owing to their emission/excitation wavelengths located in the visible light region (400–700 nm). Fortunately, recent development in fluorescent imaging was able to shift the wavelength into the second near-infrared region (NIR-II, 1000–1700 nm), which holds deeper tissue penetration and higher temporal-spatial resolution than traditional visible and NIR-I (700–900 nm) regions by minimizing auto-fluorescence and tissue scattering[35–43]. Therefore, it is expected that the integration of NIR-II fluorescent ligands into the Ru(II) metallacycle could advance the corresponding applications in biomedicine.

Herein, we report a NIR-II emissive Ru(II) metallacycle (Ru1085, $\lambda_{em} = 1085$ nm) formed via coordination-driven self-assembly using NIR-II fluorescent ligand 1 and dinuclear arene-ruthenium 2 as the building blocks (Fig. 1). Due to the clear-cut advantages of NIR excitation and NIR-II emission, Ru1085 exhibits deep tissue penetration (>6 mm) and imaging capability with high temporal-spatial resolution. Ru1085 also possesses ultrahigh photothermal conversion efficiency (PCE = 30.9%) along with reactive oxygen species (ROS) generation capability, demonstrating prospects for efficient phototherapy. Notably, Ru1085 shows desirable cellular uptake and high cytotoxicity to cisplatin-resistant A549 cells with low toxicity towards mammalian cells. Further in vivo studies successfully utilize Ru1085 in NIR-II fluorescence imaging to precisely guide and monitor the effective chemo-phototherapy on A549 tumor-bearing mice with single-dose and single laser illumination for one treatment. Collectively, this study assembles a novel Ru(II) metallacycle with long excitation/emission wavelength for NIR-II fluorescence imaging-guided and monitored improved chemo-phototherapy towards tumors.

## Results

**Design, synthesis and characterization of Ru1085.** Through molecular modeling and theoretical calculation, we introduced strong electron donor units (julolidinyl and anisole groups) and Ru(II) coordination units (thiophen-pyridine) into the acceptor aza-BODIPY skeleton to prepare NIR-II emissive ligand 1. Because of the strong intramolecular charge transfer (ICT) from donor to acceptor, the emission wavelength of ligand 1 located on the NIR-II region (Supplementary Fig. 1), which was further confirmed by theoretical calculation (Supplementary Fig. 2). The ligand 1 was confirmed by nuclear magnetic resonance (NMR) and electrospray ionization mass spectrometry (ESI-MS) (Supplementary Figs. 3–5).

Metallacycle Ru1085 was prepared by assembling NIR-II emissive ligand 1 and 0° Ru(II) acceptor 2[44] in MeOH/CHCl₃ at room temperature for 24 h (Fig. 2a). The formation of Ru1085 was then confirmed by $^1H/^{19}F$ NMR, 2D rotating frame Overhauser effect spectroscopy (ROESY) and electrospray ionization time-of-flight mass spectrometry (ESI-TOF-MS). As seen in Fig. 2b, the protons of the pyridine from Ru1085 showed diagnostic upfield shift compared with those of free ligand 1 due to the formation of the Ru–N bond. The protons of the p-cymene parts of Ru1085 also shifted upfield compared with those of free Ru(II) acceptor 2 (Fig. 2b and Supplementary Figs. 6, 7), suggesting the coordination of nitrogen atoms to Ru(II) centers. The result of $^{19}F$ NMR spectrum exhibited a single and sharp peak at −79.29 ppm, indicating the presence of free trifluoromesulfate counteranions in the same chemical environment (Supplementary Fig. 8). Besides, ROESY spectrum showed a strong cross peak between the pyridyl proton $H_{1a}$ of ligand 1 and the proton $H_{2c}$ from the p-cymene moiety of Ru1085 (Supplementary Fig. 9). Then, ESI-TOF-MS further confirmed the formation of rectangular metallacycle Ru1085 and exhibited two main peaks which were assigned to [2 + 2] assembly due to the loss of OTf⁻ counterions ($m/z = 862.46$ for [Ru1085–4OTf]⁴⁺ and $m/z = 1199.27$ for [Ru1085–3OTf]³⁺ respectively; Fig. 2c and Supplementary Fig. 10). All the peaks matched well with their theoretical distributions. Finally, the geometry optimization of Ru1085 was carried out in Gaussian 09 (Fig. 2d). The optimized geometry showed that the distance between the two Ru centers of Ru (II) acceptor 2 was 8.16 Å, which was shorter than that between the centroids of ligand 1 (21.66 Å). All above results supported the successful construction of Ru1085.

**Photophysical, photodynamic and photothermal properties, and stability studies.** Primarily, the absorption and emission spectra of Ru1085 were measured in different solvents (Supplementary Fig. 11). As shown in Fig. 3a and Supplementary Table. 1, Ru1085 exhibited a broad absorption band and centered at around 874 nm (molar extinction coefficient, $\varepsilon = 4.67 \times 10^4\ M^{-1}\ cm^{-1}$); meanwhile, the maximum emission wavelength was ~1085 nm ($\lambda_{ex} = 808$ nm) in dichloromethane (DCM). Simultaneously, the relative fluorescence quantum yield ($\Phi_f$) of Ru1085 was calculated to be 0.084% (using IR-26 as reference). The NIR-II fluorescence signals of Ru1085 was strongest under a 1000 nm long-pass (LP) filter (Fig. 3b). Hence, the 1000 nm LP filter was employed in further imaging studies. To verify the optical penetration merits, the optical penetration performance of Ru1085 was evaluated in mimic tissue (1% intralipid) with a classic photosensitizer tris(2,2′-bipyridyl)ruthenium (II) [Ru(bpy)₃Cl₂] as contrast. NIR-II fluorescence signal of Ru1085 was observed as the penetration depth increasing up to 6 mm. On the contrary, the fluorescence signal of Ru(bpy)₃Cl₂ almost became negligible with the penetration depth around 1 mm (Fig. 3c), attributed to the short absorption/emission wavelength (Fig. 3c and Supplementary Fig. 12). These results implied that

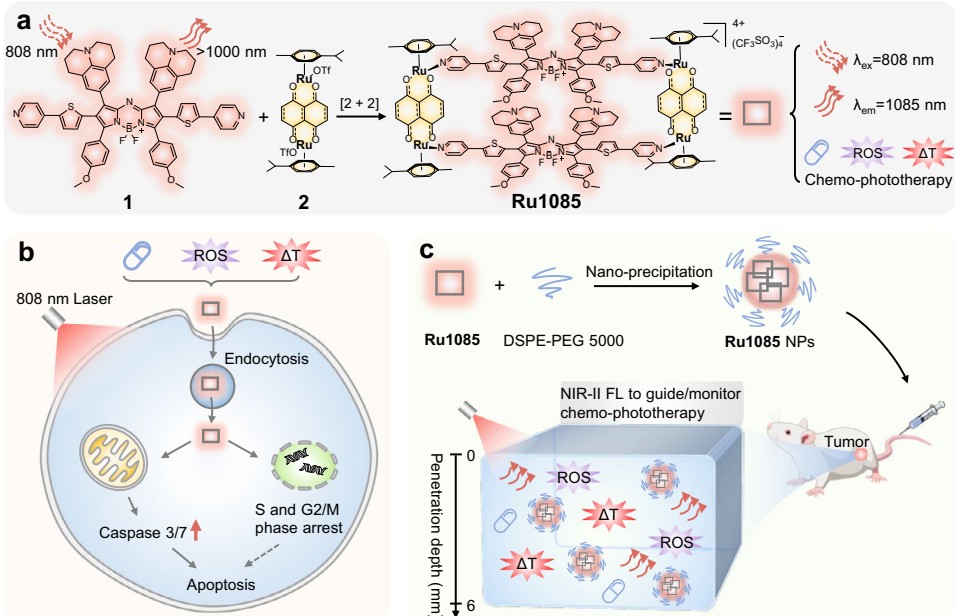

**Fig. 1 Schematic illustration of the design, antitumor mechanism, and in vivo application of Ru1085. a** The design, chemical structure, and properties of Ru1085, which is excited at 808 nm and emitted over 1000 nm with chemotherapeutic, photodynamic/photothermal properties. **b** The underlying antitumor mechanism illustrates the internalization of Ru1085 into A549 cells through endocytic pathway, and subsequently induces mitochondria-mediated apoptosis and arrested cell cycle at S and G2/M phase. **c** Ru1085 is utilized in NIR-II fluorescence imaging for guiding and monitoring chemo-phototherapy in tumor-bearing mice models.

metallacycle Ru1085 with excitation at 808 nm and emission within the NIR-II biowindow could have a potential prospect for deep tissue in vivo imaging.

Encouraged by the photophysical property of Ru1085, we further investigated singlet oxygen ($^1O_2$)/ROS production and photothermal conversion. Electron spin resonance (ESR) spectroscopy was employed to directly detect $^1O_2$ using 2,2,6,6-tetramethylpiperidine (TEMP) as the $^1O_2$ trapping agent. The time-dependent ESR spectra of Ru1085 were characterized with a 1:1:1 triplet signal (Fig. 3d). The $^1O_2$ generation capability of Ru1085 was further confirmed using 1,3-diphenyliso-benzofuran (DPBF) as indicator and the $^1O_2$ quantum yield ($\Phi_\Delta$) was measured to be 0.14 using indocyanine green (ICG) as reference (Supplementary Fig. 13)[45]. ROS generation of Ru1085 was detected with time-dependence (0–60 s) using a ROS probe 2,7-dichlorodihydrofluorescien diacetate ($H_2$-DCFH) (Fig. 3e). Besides, the photothermal behavior of Ru1085 was concentration-dependent (0–10 µM) (Supplementary Fig. 14) with high PCE (30.9%)[46] (Fig. 3f), which was superior to existing photothermal agent such as ICG (15.8%)[47]. Consequently, Ru1085 could be an excellent photodynamic and photothermal agent.

Efficient photosensitizers should possess high stability. Metallacycle Ru1085 exhibited good stability in both phosphate buffer saline (PBS) and 10% fetal bovine serum (FBS) within prolonged incubation time (7 days) (Fig. 3g and Supplementary Fig. 15). Ru1085 showed no photodegradation under 808 nm laser illumination (0.4 W cm$^{-2}$, 30 min) in PBS (Supplementary Fig. 16). Furthermore, Ru1085 exhibited excellent photothermal-stability with no significant decay of temperature in heating-cooling cycles (Fig. 3h).

**Evaluation of cell uptake and localization.** Prior to the cell experiments, the lipophilicity of Ru1085 was determined by the octanol/water partition coefficient (log $P_{o/w}$) to estimate the cell uptake efficiency. Ru1085 was more lipophilic (log $P_{o/w}$ = 1.18)

than ligand **1** (log $P_{o/w}$ = 0.54) and Ru(II) acceptor **2** (log $P_{o/w}$ = −0.79), implying the cell uptake of Ru1085 might be more effective[48]. Cell imaging tests indicated that NIR-II fluorescence signal persistently enhanced over incubation time (3–24 h) after A549 cells incubated with Ru1085 (Fig. 4a and Supplementary Fig. 17). Note that, NIR-II fluorescence signal of cells incubated with Ru1085 was much stronger than that incubated with ligand **1**. The images illustrated that metallacycle Ru1085 possessed satisfactory cell uptake and retention efficiency. Further colocalization images revealed that Ru1085 primarily localized in lysosomes with a high Pearson correlation coefficient (PCC) of 0.71 (Fig. 4b), along with partial accumulation in mitochondria and nucleus (Supplementary Fig. 18). Apart from that, laser ablation inductively coupled plasma mass spectrometry (LA-ICP-MS) was employed for clarifying the cell uptake of Ru element within single-cell-level[49]. Strong signal of $^{102}$Ru was detected in A549 cells after incubated with Ru1085 (Fig. 4c). Meanwhile, quantitative analysis was carried out by inductively coupled plasma mass spectrometry (ICP-MS). Consistent with the results of NIR-II fluorescence imaging, Ru accumulated in cells over time (3–24 h) and the majority of Ru was observed in lysosomes (Fig. 4d, e).

To identify the cellular uptake mechanism of Ru1085, cellular NIR-II fluorescence imaging was taken under different inhibition conditions. The internalization of Ru1085 was significantly inhibited at 4 °C, as well as with co-incubation of the metabolic inhibitors (2-deoxy-D-glucose and oligomycin) and endocytosis inhibitor (NH$_4$Cl)[50], revealing that the cellular uptake was energy-dependent endocytosis (Supplementary Fig. 19). Furthermore, the endocytosis inhibitors including sucrose and methyl-β-cyclodextrin (M-βCD)[51], which impair clathrin and caveolae-mediated endocytosis separately, were utilized to investigate the mechanism (Supplementary Fig. 19). The results indicated that Ru1085 was mainly internalized into cells through a clathrin- and caveolae-mediated endocytic pathway.

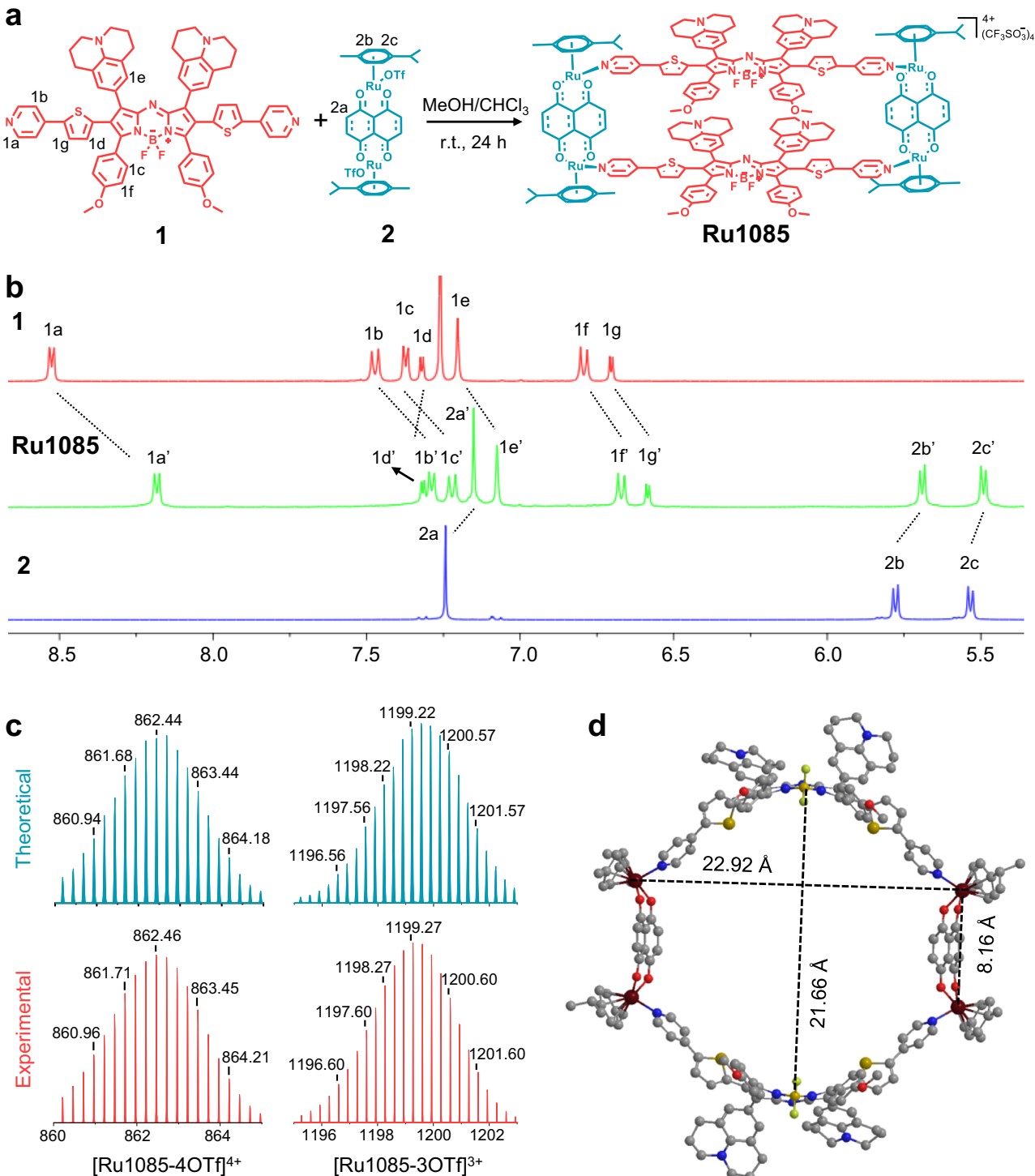

**Fig. 2 Synthesis, characterization, and theoretical calculation of Ru1085. a** Construction of ligand **1** with **2** to form **Ru1085**. **b** Partial [1]H NMR spectra of **1**, **Ru1085** and **2** (from top to bottom). **c** Theoretical (blue) and experimental (red) isotope patterns of [**Ru1085**–4OTf]$^{4+}$ (left) and [**Ru1085**–3OTf]$^{3+}$ (right) for ESI-TOF-MS. **d** Molecular model of **Ru1085** optimized by B3LYP molecular orbital approach.

**In vitro antitumor activity and mechanism**. The antitumor activity of was assessed against A549 (lung), Hela (cervix) and HepG2 (liver) tumor cell lines. Ru(bpy)$_3$Cl$_2$, cisplatin and 5-ALA (5-aminolevulinic acid, clinically PDT agent) were chosen as comparisons (Table 1). **Ru1085** presented tremendous antitumor capability with high cytotoxicity values in above cell lines. With 808 nm laser illumination, enhanced cytotoxicity of **Ru1085** was found, especially in A549 cells (IC$_{50, \text{light}}$ = 4.5 μM). The

phototoxic index (PI, defined as IC$_{50, \text{dark}}$/IC$_{50, \text{light}}$) of **Ru1085** in A549 cells could reach the highest value among examined cell lines. Hence, A549 cell line was selected for further exploration. Low or almost no cytotoxicity was found for 5-ALA and Ru(bpy)$_3$Cl$_2$ (IC$_{50}$ range from 87.4 to >300 μM).

Considering the hypoxic environment of solid tumor, the antitumor activity was estimated under hypoxia (1% O$_2$). Excitingly, no obvious decline in antitumor efficiency of

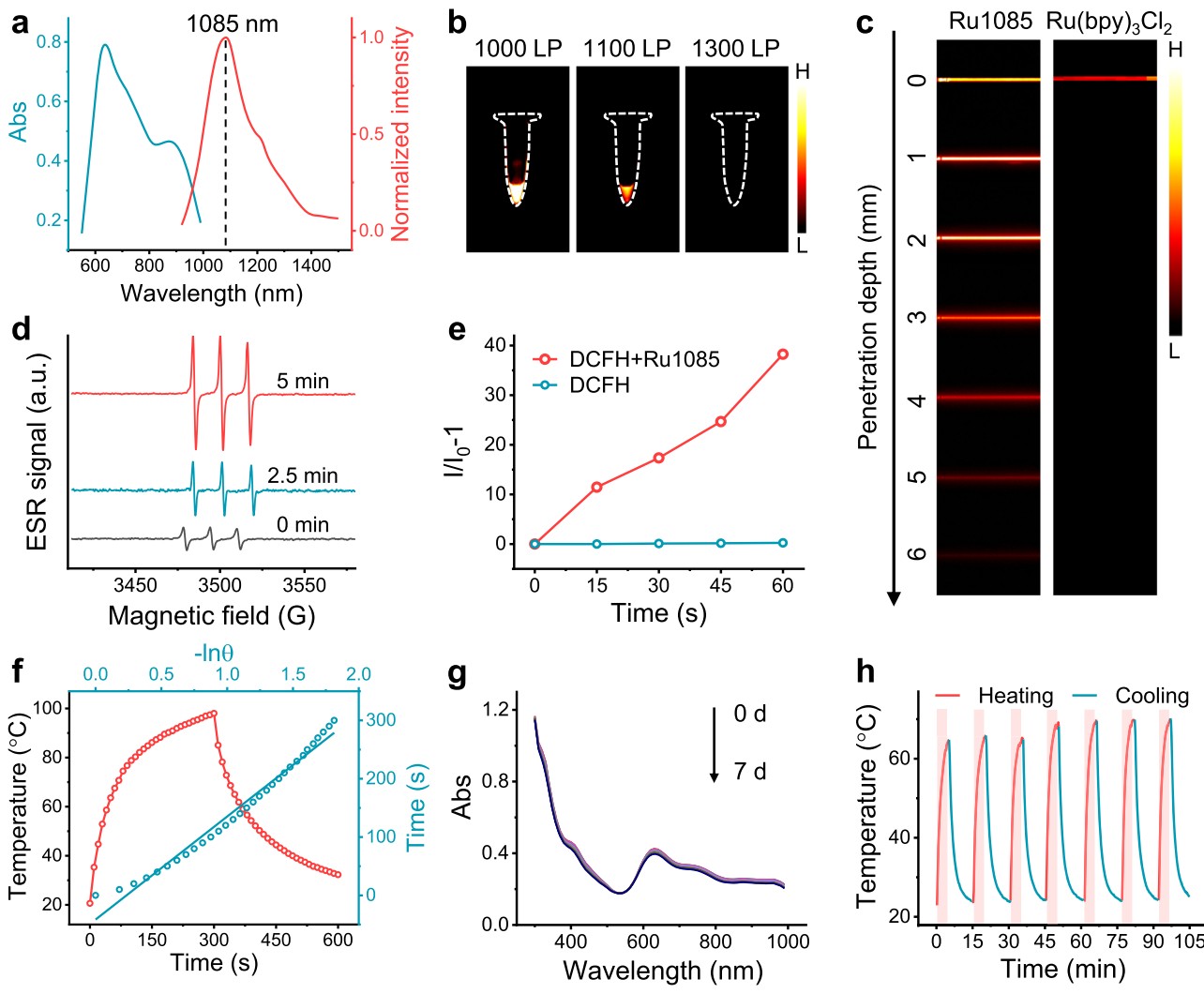

**Fig. 3 In vitro photophysical, photodynamic, photothermal properties, and stability of Ru1085. a** Absorption and normalized emission spectra ($\lambda_{ex} = 808$ nm) of **Ru1085** in DCM. **b** NIR-II fluorescence images of **Ru1085** under different long-pass filters. **c** Fluorescence images of **Ru1085** and Ru(bpy)$_3$Cl$_2$ encapsulated in capillaries and immersed at different depths in 1% intralipid. **d** Detection of $^1$O$_2$ analyzed by ESR measurement. **e** ROS generation of **Ru1085** (20 μM) illuminated by 808 nm laser (0.8 W cm$^{-2}$) using H$_2$-DCFH as indicator. **f** Monitored temperature profile (red line) of **Ru1085** illuminated for 300 s and followed by natural cooling, and linear time data versus $-\ln\theta$ (blue line) from the cooling period. **g** The stability tests of **Ru1085** incubated in 10% FBS for 7 days. **h** Photothermal stability of **Ru1085** (10 μM) illuminated by 808 nm laser (0.8 W cm$^{-2}$) for seven repeating cycles of heating-cooling.

**Ru1085** was observed when cells moved from normoxia (IC$_{50,\ light}$ = 4.5 μM) to hypoxia (4.9 μM). We speculated that the photocytotoxicity of **Ru1085** was not primarily dependent on $^1$O$_2$. ROS scavenging tests proved the generation of very minor $^1$O$_2$, and the produced ROS types mainly included hydroxyl radical (OH$^\bullet$), hydrogen peroxide (H$_2$O$_2$) and superoxide radical (O$_2$$^{\bullet-}$) (Supplementary Fig. 20). On the contrary, the PDT agent 5-ALA presented much lower photocytotoxicity in hypoxia (IC$_{50,\ light}$ > 300 μM) than that in normoxia (87.4 μM), indicating the antitumor efficiency of 5-ALA would dramatically reduce in solid tumor. Meanwhile, the activity of **Ru1085** against drug-resistant tumor cells was investigated in cisplatin-resistant A549 cells (A549cisR). The resistant factor (RF) was defined as IC$_{50}$ in A549/IC$_{50}$ in A549cisR. Promisingly, **Ru1085** exhibited prominent antitumor activity in overcoming drug resistance with a high RF of 2.2, which was ~4-fold of that of cisplatin (RF = 0.5). Attractively, **Ru1085** showed desirable selectivity in normal cells (16HBE cell line) with a high selectivity index (SI = 3.4), defined as IC$_{50}$ in 16HBE/IC$_{50}$ in A549, while cisplatin was even more

toxic in normal cells (0.3). Therefore, these results showed that **Ru1085** presented prominent antitumor capability and could be a candidate for further in vivo antitumor practice.

Inspired by the admirable cytotoxicity against malignant cells, the possible antitumor mechanism of **Ru1085** was then investigated. First, considering **Ru1085** mainly localized in lysosomes, the integrity of lysosomes was confirmed using acridine orange (AO) staining. Once lysosomes were damaged, the red fluorescence of AO would reduce. As depicted in Fig. 5a and Supplementary Fig. 21, the red fluorescence of AO almost disappeared, indicating that the lysosomes were dysfunctional after treated with **Ru1085** and **Ru1085** along with laser illumination. Moreover, **Ru1085** also partially localized in mitochondria. Thus, 5,5′,6,6′-tetrachloro-1,1′,3,3′-tetraethylben-zimidazolylcarbocyanine iodide (JC-1), an indicator of the mitochondrial membrane potential (MMP), was employed to monitor the change of MMP. When MMP was reduced, the fluorescence of JC-1 could change from red to green. The green fluorescence indicated that **Ru1085** could significantly induce

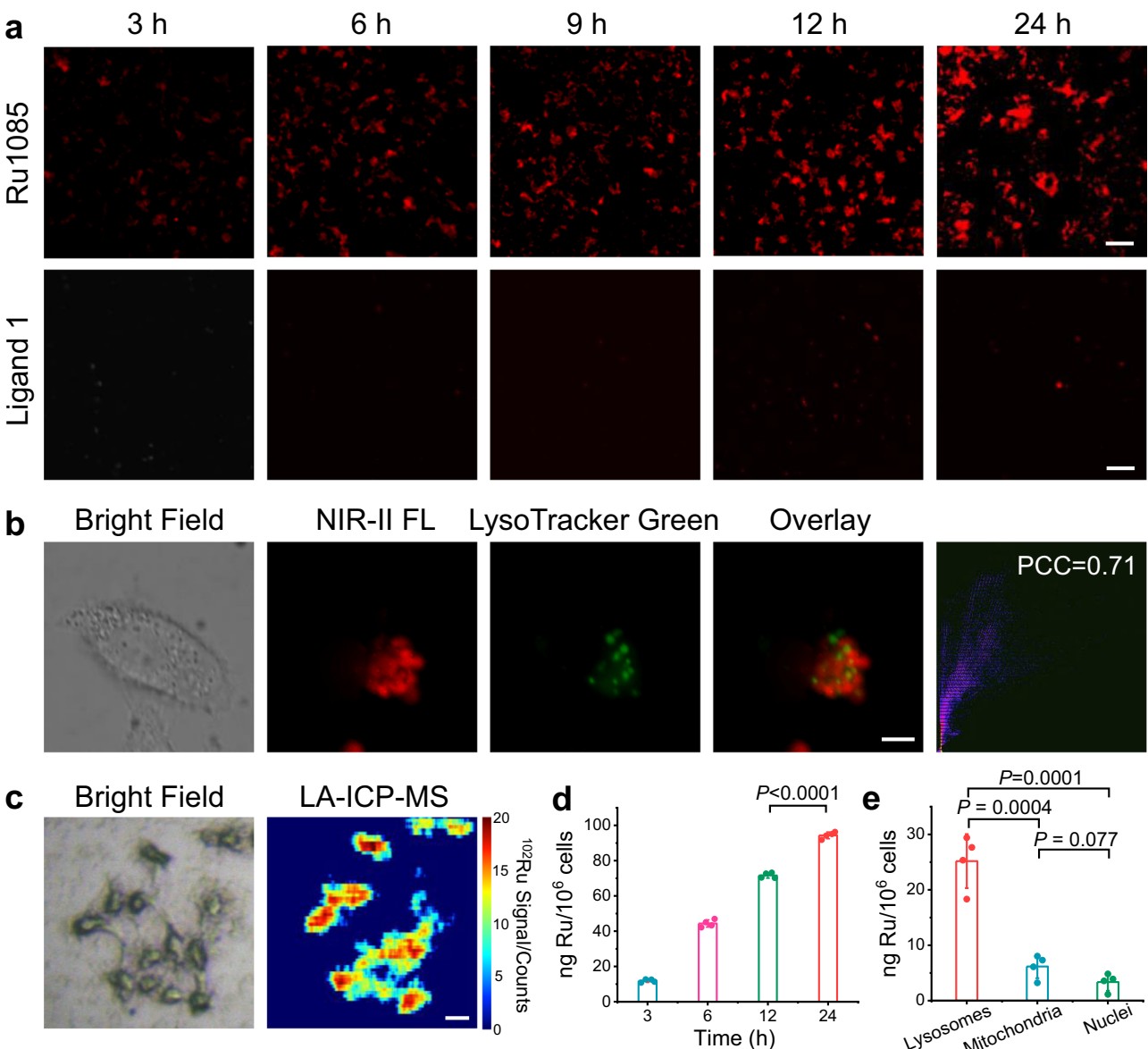

**Fig. 4 Cellular uptake and localization of Ru1085 in A549 cells. a** NIR-II fluorescence images of cells incubated with **Ru1085** (10 µM) and ligand **1** (20 µM) over time. Scale bar, 50 µm. **b** Colocalization assay of **Ru1085** (10 µM) using LysoTracker Green as lysosomal dye. The PCC was calculated to be 0.71. Scale bar, 5 µm. **c** LA-ICP-MS imaging of $^{102}$Ru in cells after incubated with **Ru1085** (5 µM). Scale bar, 20 µm. ICP-MS results of **d** intracellular Ru amount with time-dependence and **e** Ru localization after incubated with **Ru1085** (5 µM) for 6 h. Data were presented as mean ± s.d. ($n = 4$). Statistical differences were analyzed by Student's two-sided $t$-test. Source data are provided in Source Data file.

mitochondrial depolarization and the laser illumination further aggravated mitochondrial damage (Fig. 5b and Supplementary Fig. 22). Mitochondria are generally sensitive to toxic ROS, and oxidative stresses induced by ROS can cause mitochondrial membrane depolarization[52]. Therefore, ROS production was tested by the staining with H₂-DCFH, which could be deacetylated in cells and oxidized by ROS into DCF emitting green fluorescence. The DCF fluorescence remarkably increased after treated with **Ru1085** along with laser illumination (Fig. 5c), indicating ROS was generated in cells. Mitochondria damage often triggers the downstream caspase cascade activation and apoptosis. Hence, we evaluated the expression of caspase 3/7 and **Ru1085** efficiently activated caspase 3/7, especially in **Ru1085** along with laser group (Fig. 5d). In addition, caspase 1, as a member of the cysteine protease family, facilitates the activations of caspase 3/7[53]. Furthermore, caspase 1 activity tests showed that

**Ru1085** could activate caspase 1 (Fig. 5e). Given that active caspase 3/7 is an effector in apoptotic pathways, cell apoptosis was then investigated using Annexin V-FITC/propidium iodide (PI) double-staining. As shown in Fig. 5f, **Ru1085** mainly induced early apoptosis (AV + /PI − ) and **Ru1085** along with laser illumination mainly caused late apoptosis/necrosis (AV + /PI + ). Due to the nuclear accumulation of **Ru1085**, the influence for nucleus was further evaluated by analyzing cell cycle distribution. The cell cycle was mainly arrested at S and G2/M phase, suggesting the suppression of DNA replication and mitosis (Fig. 5g, h). In conclusion, the underlying antitumor mechanism of **Ru1085** was attributed to mitochondria-mediated apoptosis and cell cycle arrest at S and G2/M phase.

In addition to inhibiting malignant cell proliferation, the antimetastatic activity of **Ru1085** was explored because some Ru(II) complexes exhibited suppressing tumor metastasis in

**Table 1 IC$_{50}$ value of Ru1085, Ru(bpy)$_3$Cl$_2$, cisplatin and 5-ALA against various cell lines by MTT assay.**

| Condition | Cell line | Dark/light[a] | IC$_{50}$ (µM) | | | | PI[b] |
|---|---|---|---|---|---|---|---|
| | | | Ru1085 | Ru(bpy)$_3$Cl$_2$ | Cisplatin | 5-ALA | |
| Normoxia | A549 | Dark | 11.6 ± 1.8 | >300 | 21.5 ± 0.1 | >300 | 2.5 |
| | | Light | 4.5 ± 0.5 | >300 | - | 87.4 ± 1.1 | |
| | Hela | Dark | 10.0 ± 2.9 | >300 | 21.2 ± 2.0 | >300 | 1.7 |
| | | Light | 5.9 ± 1.9 | >300 | - | 168.0 ± 27.0 | |
| | HepG2 | Dark | 21.2 ± 1.3 | >300 | 9.1 ± 1.4 | >300 | 1.5 |
| | | Light | 13.5 ± 4.2 | >300 | - | 101.9 ± 13 .9 | |
| | A549cisR | Dark | 5.3 ± 0.3 | >300 | 44.7 ± 3.6 | >300 | 1.4 |
| | | Light | 3.9 ± 0.4 | >300 | - | 268.8 ± 16.0 | |
| | | RF[c] | 2.2 | - | 0.5 | - | |
| | 16HBE | Dark | 39.4 ± 8.4 | >300 | 5.5 ± 0.3 | >300 | 1.1 |
| | | Light | 37.1 ± 12.9 | >300 | - | | |
| | | SI[d] | 3.4 | - | 0.3 | - | |
| Hypoxia | A549 | Dark | 8.1 ± 1.4 | >300 | 37.6 ± 1.2 | >300 | 1.7 |
| | | Light | 4.9 ± 1.2 | >300 | - | >300 | |

[a]The light source used for **Ru1085** was 808 nm laser and the light source used for Ru(bpy)$_3$Cl$_2$ and 5-ALA was 450 nm LED.
[b]PI (phototoxic index) = IC$_{50, dark}$ treated with **Ru1085**/IC$_{50, light}$ treated with **Ru1085**.
[c]RF (resistant factor) = IC$_{50, dark}$ in A549/IC$_{50, dark}$ in A549cisR.
[d]SI (selectivity index) = IC$_{50, dark}$ in 16HBE/IC$_{50, dark}$ in A549.

previous reports[54,55]. The wound-healing assay was performed to evaluate the anti-migration efficiency of **Ru1085** (Supplementary Fig. 23). Attractively, only after treated with **Ru1085** and **Ru1085** along with laser illumination, the migration of cells was efficiently suppressed (wound closure ratio <7%). Afterwards, a transwell invasion assay was utilized to assess the anti-invasion potential of **Ru1085** (Supplementary Fig. 24). The diminished invasion was observed after treated with Ru1085 and Ru1085 along with laser illumination with an invasion ratio of 45 and 25%, respectively. To sum up, the above data indicated that **Ru1085** also significantly inhibited the migration and invasion of tumor cells.

**In vivo NIR-II fluorescence imaging-guided and monitored tumor chemo-phototherapy.** With above promising in vitro antitumor capability, the in vivo application of **Ru1085** was further explored. To improve the tumor-targeting ability, **Ru1085** was encapsulated into DSPE-PEG5000 (encapsulation rate *ca.* 31%, Supplementary Fig. 25). After encapsulation, the absorption and emission spectra of **Ru1085** NPs had minimal shift compared with **Ru1085** (Supplementary Fig. 26). PCE of **Ru1085** NPs was calculated to be 36.7% (Supplementary Fig. 27). Dynamic light scattering (DLS) results suggested that the average diameter of **Ru1085** NPs was ~220 nm in water, and the size obtained from transmission electron microscopy (TEM) imaging was ~170 nm (Supplementary Fig. 28). To determine the stability of **Ru1085** NPs, the optical and size profiles in physiological condition were carried out. After incubated in 10% FBS or whole blood and storage for a whole week, the absorption spectra of **Ru1085** NPs showed negligible change (Supplementary Fig. 29). Meanwhile, the size distribution had minimum change after incubated in PBS or 10% FBS (Supplementary Fig. 30). With continuous laser illumination, no significant change was observed in absorbance (Supplementary Fig. 31). Prior to intravenous injection, the biosafety of **Ru1085** NPs was investigated by performing hemolysis assay (Supplementary Fig. 32). **Ru1085** NPs exhibited no hemolysis at various concentration (5–120 µM), which can be attributed to the encapsulation of **Ru1085** by liposome. These data verified the stability and biosafety of **Ru1085** NPs, which enabled the further in vivo application.

To estimate the in vivo NIR-II fluorescence performance of **Ru1085** NPs, we carried out NIR-II fluorescence imaging of the regional vascular system of hindlimb. The arteries and veins could be clearly visualized from the background skin tissue with excellent signal to background ratio (SBR = 13.6) and ideal spatial resolution (narrow full width at half-maximum (FWHM) = 424 µm) (Fig. 6a, b). In contrast, the vessels could not be distinguished from skin using Ru(bpy)$_3$Cl$_2$ NPs as probe, confirming the drawbacks of short wavelength (Supplementary Fig. 33). In light of the successful in vivo NIR-II fluorescence imaging of **Ru1085** NPs, A549 tumor-bearing mice models were established. We employed ICP-MS to evaluate pharmacokinetics by quantifying Ru amounts in plasma after injection and **Ru1085** NPs exhibited a prolonged circulation time in blood (Fig. 6c). Meanwhile, the NIR-II fluorescence imaging was collected at various time points post-injection, and the signal in tumor was gradually enhanced and reached the highest value at 24 h post-injection of **Ru1085** NPs with a high SBR of 6.6 (Fig. 6d and Supplementary Fig. 34). Tumors and other organs were collected at 24 h post-injection for analysis of the NIR-II fluorescence intensity, illustrating the successful tumor accumulation (Supplementary Fig. 35).

Under the guidance of NIR-II fluorescence imaging, the synergistic therapeutic effect was then explored. A459 xenograft tumor models were randomly divided into five groups (n = 5/ group) and intravenously injected with **Ru1085** NPs (1 mg Ru/kg), cisplatin (1 mg Pt/kg), or PBS (10 mM). The tumors were subjected with or without 808 nm laser illumination at 24 h post-injection. The photothermal images and temperature elevation were recorded by an infrared thermal camera (Fig. 6e and Supplementary Fig. 36). The temperature of tumor in **Ru1085** NPs along with laser illumination group reached 54.1 °C. Besides, the ROS generation in tumors was tested using dihydroethidium (DHE, a ROS indicator) staining. Strong fluorescence was observed in **Ru1085** NPs along with laser illumination group, indicating ROS production in tumor after phototherapy (Fig. 6f).

Furthermore, NIR-II fluorescence imaging was performed for real-time monitoring the tumor lesion and depicting tumor margin with high SBR (SBR >4, 12–120 h) during the chemo-phototherapy period (Fig. 6g and Supplementary Fig. 37). To compare therapeutic effect of different groups, the tumor volume was recorded every other day (Fig. 6h and Supplementary Fig. 38). In **Ru1085** NPs and cisplatin group, the tumor growth was under suppression at the beginning of chemotherapy, but severely recurred on the eighth day, suggesting that the single

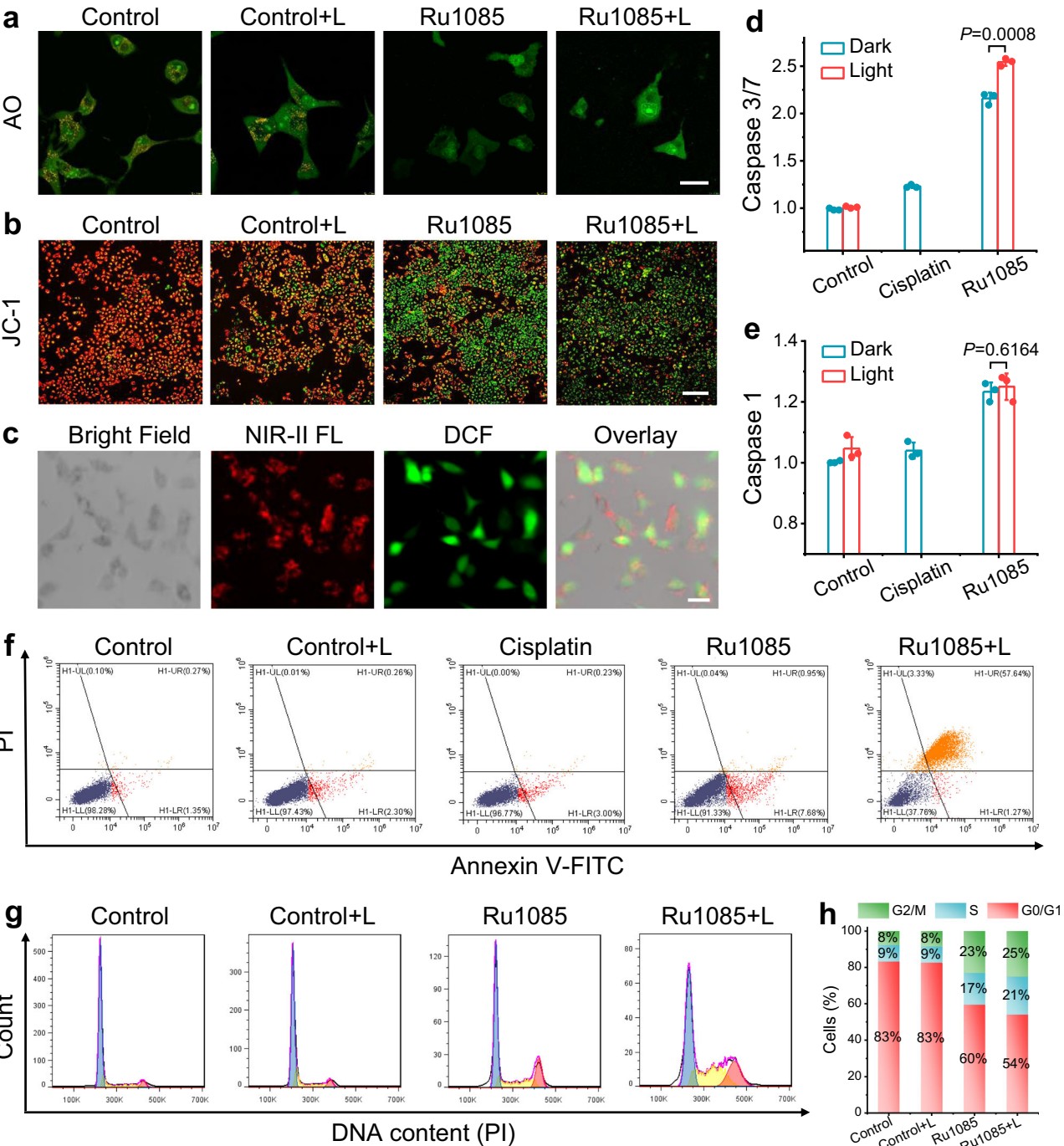

**Fig. 5 Antitumor mechanism of Ru1085 in A549 cells.** After incubated with **Ru1085** (10 μM) or serum-free medium and treated with or without 808 nm laser illumination (0.8 W cm$^{-2}$, 5 min), cells were taken **a** AO and **b** JC-1 staining. Scale bars, **a**: 20 μm; **b**: 300 μm. **c** H$_2$-DCFH staining of cells treated with **Ru1085** along with laser illumination. Scale bar, 20 μm. Relative caspase 3/7 (**d**) and caspase 1 (**e**) activity after various treatments ($n = 3$). **f** Flow cytometer results of Annexin V-FITC/PI double-stained cells after various treatments. **g** Cell cycle analysis of cells after various treatments. **h** Histogram depicting of the cell population distribution in cell cycle phase in (**g**). Data were presented as mean ± s.d. ($n = 3$). Statistical differences were analyzed by Student's two-sided $t$-test. Source data are provided in Source Data file.

chemotherapy could not eradicate tumor. Only in chemo-phototherapy group which was administrated with **Ru1085** NPs along with laser illumination, the tumor was successfully suppressed and almost no recurrence was detected. Histological staining of the tumors in each group showed that the severe cellular necrosis was only found in chemo-phototherapy group (Supplementary Fig. 39), supporting the therapeutic effect as mentioned. Negligible body weight loss was found after

therapeutic period (Fig. 6i). Notably, the body weight of cisplatin group obviously declined (from $18.5 ± 0.6$ g to $17.7 ± 0.9$ g) during the first four days, indicating the potential toxicity of cisplatin. Further histological staining of major organs and blood biochemical parameters revealed that no tissue damage and no abnormality in hepatorenal function after administration of **Ru1085** NPs (Supplementary Figs. 40, 41), which suggested negligible long-term systemic toxicity was introduced by **Ru1085**

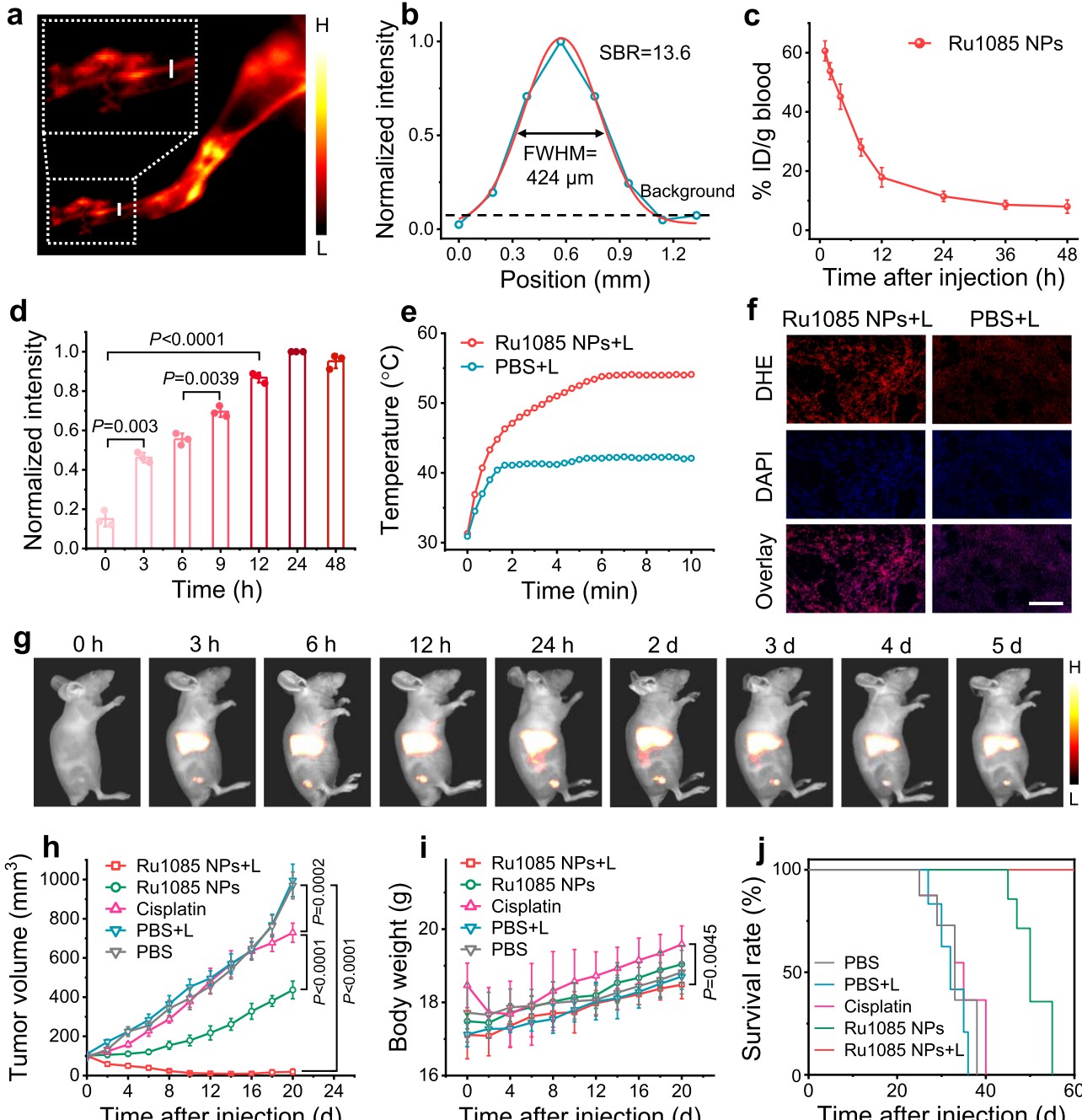

**Fig. 6 In vivo NIR-II fluorescence imaging and chemo-phototherapeutic effect of Ru1085 NPs in A549 tumor-bearing mice. a** NIR-II fluorescence images of hindlimb vessels after intravenous injection of **Ru1085** NPs. **b** NIR-II fluorescence intensity profiles (blue line) and Gaussian fit (red line) along the white full line in (**a**). **c** Blood retention and **d** analysis of NIR-II fluorescence signals in tumor regions after injection of **Ru1085** NPs. Data were presented as mean ± s.d. ($n = 3$ independent mice). **e** Temperature change in tumor region with 808 nm laser illumination (0.8 W cm$^{-2}$, 10 min) at 24 h post-injection of **Ru1085** NPs or PBS. **f** DHE and DAPI staining of tumor slices. Scale bar, 400 μm. **g** In vivo NIR-II fluorescence images of tumor models after injection of **Ru1085** NPs. **h** Tumor inhibitory effect and **i** mice body weight of tumor models within 20 days of treatment. **j** Kaplan–Meier survival plots for various treated mice (**Ru1085** NPs dose: 1 mg Ru/kg, cisplatin dose: 1 mg Pt/kg, laser treatments: 0.8 W cm$^{-2}$ for 10 min). Data were presented as mean ± s.d. ($n = 5$ independent mice). Statistical differences were analyzed by Student's two-sided $t$-test. Source data are provided in Source Data file.

NPs. To further assess the therapeutic effect, the life-time of mice models after various treatments was recorded. The chemo-phototherapy could significantly prolong the survival time to over 60 days (Fig. 6j). All these data specified that **Ru1085** NPs inherent with biosafety successfully accomplished high-resolution NIR-II fluorescence imaging, guided chemo-phototherapy with enhanced performance, and monitored the therapeutic response.

## Discussion

Most existing Ru-based agents suffered from short excitation/emission wavelength and the corresponding excitation light failed to penetrate deep into the tissues in phototherapy. Herein, we successfully designed a metallacycle **Ru1085** which was excited at 808 nm and emitted over 1000 nm. The 808 nm laser can penetrate deeper tissues (~6 mm) compared with the short-wavelength

light source (<600 nm) that was employed for Ru complexes in previous reports[12,56].

**Ru1085** exhibited much more efficient uptake in cells than ligand **1**, which could be attributed to the higher lipophilicity[44], positive charge[57], and characteristic ring tension[58] of metallacycle. Further studies suggested the endocytic uptake mechanism and **Ru1085** mainly localized in lysosomes along with partially enriched in mitochondria and nucleus. The positive-charge and lipophilicity of **Ru1085** could resulted in the localization in mitochondria and nucleus[48,59], which facilitated toxic ROS inducing cell death considering mitochondria and nucleus are sensitive to ROS[60].

Afterwards, the in vitro antitumor performance of **Ru1085** was evaluated in details. First, **Ru1085** presented higher dark/photo-cytotoxicity in various tumor cell lines than Ru(bpy)$_3$Cl$_2$ and 5-ALA, especially in A549 cell line. Second, compared with the significant photocytotoxicity decrease of clinical photosensitizer 5-ALA during the transfer of cells from normoxia to hypoxia, the antitumor efficiency of **Ru1085** had almost no change regardless of cells in normoxia or in hypoxia. Last, **Ru1085** was capable of overcoming cisplatin resistance and possessed selectivity between cancer cells and normal cells. It is anticipated that further improvements could be realized by optimizing the PI value based on the strategy of increasing photocytotoxicity and reducing dark cytotoxicity simultaneously. For enhancing SI value, special peptides[61] could be introduced into the skeleton to increase the specific uptake of tumor cells. Mechanistic studies revealed that **Ru1085** induced mitochondria-mediated apoptosis as well as S and G2/M phase cell cycle arrest. The exploration of antitumor mechanism of **Ru1085** helps us understand the cell death process of the synergistic therapy and provides first-hand information for rationally designing Ru-based metallacycle.

Further in vivo studies demonstrated the advantages of long excitation/emission wavelength of **Ru1085**. Notably, the precise profile of blood vessels with high SBR (over 10) and desirable spatial resolution (FWHM was at micron-level) was accomplished by the NIR-II fluorescence imaging. In xenografted tumor models, the NIR-II fluorescence imaging was able to guide the synergistic therapy and delineate the tumor change in real-time with precise tumor margin during the therapy period, which also provided essential imaging data for long-term assessing the therapeutic effect. Herein, through one treatment of single dose of **Ru1085** NPs along with single laser illumination, the theranostics was achieved and the synergistic therapy was more efficient than single therapy. Therefore, the construction of long-wavelength emissive metal-based agents could enhance optical penetration for improving phototherapeutic efficacy in deep and/or solid tumors. Besides, taking the intrinsic advantage of long-wavelength emission, they could be utilized as a universal platform for visualizing the delivery, targeting, pharmacokinetics, and distribution through fluorescence imaging. Finally, such a system could provide real-time feedback to the treatment, and facilitate the clinic translation of metal agents in synergistic chemo-phototherapy in the future.

In summary, we have constructed a novel Ru(II) metallacycle (**Ru1085**) emitting over 1000 nm and achieved precise NIR-II fluorescence imaging-guided and monitored chemo-phototherapy. By introducing aza-BODIPY (**1**) as an NIR-II fluorescence emitter into the Ru(II) metallacycle skeleton, the emission peak of **Ru1085** was shifted into NIR-II biowindow compared with traditional Ru complexes, and the optical penetration depth reached up to 6 mm, facilitating NIR-II fluorescence imaging with high spatial-temporal resolution and efficient phototherapy. The anticancer capability of **Ru1085** was prominent in a cisplatin-resistant cell line with low toxicity on normal cells. Through one treatment, **Ru1085** could be successfully utilized in NIR-II fluorescence imaging-guided chemo-

phototherapy and monitored long-term therapeutic response in tumor models. This work proposed a novel strategy to develop an emitted over 1000 nm metallacycle, which provided promising opportunities for metal-based agents applied in biomedicine.

## Methods

**Synthesis and characterization of Ru1085.** To synthesize **Ru1085**, **1** (5.20 mg, 0.00488 mmol) and **2** (4.82 mg, 0.00488 mmol) were dissolved in a mixture of MeOH (4 mL) and CHCl$_3$ (4 mL) in a glass vial. After stirring at room temperature for 24 h, the assembled product was concentrated by rotary evaporation, precipitated by addition of diethyl ether, and washed three times with diethyl ether followed by dried under vacuum to afford green solids of **Ru1085** (7.30 mg, yield 72%). Structural characterization was determined by $^1$H NMR, $^{19}$F NMR, 2D rotating frame Overhauser effect spectroscopy (ROESY), and electrospray ionization time-of-flight mass spectrometry (ESI-TOF-MS).

**Calculation of NIR-II fluorescence quantum yield (Φ$_f$).** The NIR-II fluorescence quantum yield (Φ$_f$) was measured using a relative strategy. The Φ$_f$ was calculated using IR-26 (Φ$_f$ = 0.1%) as reference according to the following formula:

$$\Phi_{f(X)} = \Phi_{f(ref)} \times \frac{S_X}{S_{ref}} \times \left(\frac{n_{ref}}{n_X}\right)^2 \quad (1)$$

where subscripts X and ref designate **Ru1085** and IR-26, respectively. S stands for the slope obtained by linear fitting of the integrated emission spectrum against the absorbance at 808 nm, and $n$ stands for the refractive indices of their respective solvents.

**Tissue phantom imaging study.** 1% intralipid was used to simulate tissue due to its similar scattering features. Glass capillaries were filled with **Ru1085** and Ru(bpy)$_3$Cl$_2$, and then covered with various volumes of intralipid in a dish for imaging. The fluorescence images were then obtained by using 808 nm laser illumination for **Ru1085**, and 460 nm laser illumination for Ru(bpy)$_3$Cl$_2$.

**Assessment of $^1$O$_2$ generation.** ESR analysis was performed to monitor the generation of $^1$O$_2$. ESR spectra of **Ru1085** solution (5 μM) containing TEMP (300 mM) were obtained by using 808 nm laser illumination (0.8 W cm$^{-2}$) with various illumination time.

The evaluation of $^1$O$_2$ quantum yield (Φ$_\Delta$) used DPBF as probe via UV−Vis spectroscopy. The working samples containing DPBF (20 μg mL$^{-1}$) and **Ru1085** solution (20 μM) or ICG (20 μM) were irradiated with 808 nm laser (0.8 W cm$^{-2}$) for various time. The Φ$_\Delta$ was calculated using ICG (Φ$_\Delta$ = 0.2) as reference according to the following formula:

$$\Phi_{\Delta(X)} = \Phi_{\Delta(ref)} \times \frac{S_X}{S_{ref}} \times \frac{F_{ref}}{F_X} \quad (2)$$

where subscripts X and ref designate **Ru1085** and ICG, respectively. S stands for the slope of plot of the absorbance of DPBF (at 405 nm), and F stands for the absorption correction factor ($F = 1-10^{-OD808}$).

**Assessment of ROS generation.** H$_2$-DCFH (1.0 mM, 0.8 mL) in DMSO was mixed with NaOH (0.01 M, 2 mL) to deacetylate into DCFH. Added the prepared DCFH (20 μM) into **Ru1085** solution (20 μM) and then irradiated with 808 nm laser for 0, 30, 60, 90, and 120 s. The fluorescent spectra of DCF ($\lambda_{ex}$ = 488 nm, $\lambda_{em}$ = 525 nm) were recorded.

**Photothermal properties and the photothermal conversion efficiency (PCE) of Ru1085.** **Ru1085** solution (0, 2.5, 5, 7.5, and 10 μM) was irradiated with 808 nm laser (0.8 W cm$^{-2}$) for 5 min, and was recorded by an infrared thermal imaging camera. The photothermal conversion efficiency (PCE) of **Ru1085** was calculated according to the following formula, in which $T_{max}$ (or $T_{sur}$) is the equilibrium temperature (or ambient temperature), $I$ is the incident laser power ($I = 0.8$ W cm$^{-2}$), $A_{808}$ is the absorbance at 808 nm, and $\tau_s$ is the system time constant of the sample.

$$PCE = \frac{hs(T_{max} - T_{sur}) - Q_0}{I(1 - 10^{-A_{808}})} \quad (3)$$

$$hs = \frac{\sum_i m_i C_{p,i}}{\tau_s} \quad (4)$$

$$\tau_s = \frac{t}{-ln\vartheta} = \frac{t}{-\ln\left(\frac{T - T_{sur}}{T_{max} - T_{sur}}\right)} \quad (5)$$

**Stability tests of Ru1085.** For chemical-stability tests, **Ru1085** (50 μM) was incubated in PBS or 10% FBS and stored for a whole week. The UV-Vis absorption of **Ru1085** was measured every day. The photostability of **Ru1085** was investigated

by recording the absorption spectra, for which the UV-Vis absorption of **Ru1085** (50 μM) was measured after 808 nm laser illumination (0.4 W cm$^{-2}$) for various time (0, 10, 20, 30 min). For photothermal stability tests, **Ru1085** (10 μM) was illuminated with 808 nm laser (0.8 W cm$^{-2}$) for 5 min and then naturally cooled for 10 min. The temperatures of seven heating-cooling cycles were recorded using an infrared thermal imaging camera.

**Measurement of octanol/water partition coefficient (log $P_{o/w}$).** Utilizing the "shake-flask" method to detect the distribution coefficient between water and octanol phase. The used phases were saturated in each other at first. Then **Ru1085** and ligand **1** were dissolved in phase A. This solution was subsequently mixed with an equal volume of phase B for 24 h using a mixer. Phase A was then carefully separated from phase B. The concentrations of **Ru1085** and **1** were determined by UV-Vis spectroscopy using the extinction coefficients of the complexes in water saturated with octanol. The log $P_{o/w}$ value was calculated according to the following formula:

$$\log P_{o/w} = \log c[X]_{oct} - \log c[X]_{PBS} \qquad (6)$$

**Cellular uptake by fluorescence imaging.** A549 cells were seeded on confocal dishes (~5 × 10$^4$ cells/dish). After incubated with **Ru1085** (10 μM) and ligand **1** (20 μM) for various time, A549 cells were washed three times and subjected for imaging by utilizing an NIR-II fluorescence microscope ($\lambda_{ex}$ = 808 nm, $\lambda_{em}$ = 1000–1200 nm).

**Intracellular colocalization by fluorescence imaging.** A549 cells were incubated with **Ru1085** (10 μM) for 6 h and then further incubated with LysoTracker Green (200 nM, 1 h), MitoTracker Deep Red (75 nM, 45 min), and Hoechst 33342 (5 μg mL$^{-1}$, 20 min), respectively. After washed three times by PBS, cells were visualized by a fluorescence microscope. Fluorescence images were collected in the following channels: LysoTracker Green ($\lambda_{ex}$ = 488 nm, $\lambda_{em}$ = 520–550 nm); Mito-Tracker Deep Red ($\lambda_{ex}$ = 644 nm, $\lambda_{em}$ = 665–700 nm); Hoechst 33342 ($\lambda_{ex}$ = 405 nm, $\lambda_{em}$ = 430–460 nm); **Ru1085** ($\lambda_{ex}$ = 808 nm, $\lambda_{em}$ = 1000–1200 nm). Pearson correlation coefficient was quantified using ImageJ.

**Elemental imaging by LA-ICP-MS.** A549 cells were seeded in 24-well culture plates with cell climbing slices for overnight and then incubated with **Ru1085** (5 μM) for 6 h. After washed by PBS w/o Ca/Mg, cells were fixed with 70% cold alcohol solution and washed by water. Then, cells were allowed to adhere on slides to prepare LA-ICP-MS test. A laser spot size of 3 μm diameter, 10 μm s$^{-1}$ scan speed, 100 Hz repetition frequency, and laser fluence of ~3 J cm$^{-2}$ were utilized to perform the test. ICP-MS parameters were as follows: radio frequency power of 1500 W, nebulizer gas flow of 1.25 L min$^{-1}$, auxiliary gas flow of 1.2 L min$^{-1}$, and plasma gas flow of 15 L min$^{-1}$. The monitored isotope $^{102}$Ru was measured in counting mode. Images integration was performed by the software Igor-based Iolite V3.6.

**Cellular uptake and localization by ICP-MS.** For cellular uptake tests, A549 cells were incubated with **Ru1085** (5 μM) for various time (3, 6, 12, and 24 h) and then harvested for accurate cell counting. After digested using 60% HNO$_3$, the samples were diluted to make the concentration of HNO$_3$ to 2%. The Ru content was determined by ICP-MS associated with the total cell numbers.

For cellular localization tests, A549 cells were incubated with **Ru1085** (5 μM) for 6 h and then harvested for accurate cell counting and equally separating into three parts. Each part was proceeded according to the protocol of lysosomal/mitochondria/nucleus extraction kit. Following process was conducted as above mentioned to determine the Ru content in various organelles.

**Mechanism studies of cellular uptake.** A549 cells were seeded on confocal dishes (~5 × 10$^4$ cells/dish). Cells were divided into six groups and pretreated with (1) **Ru1085** (10 μM) at 37 °C; (2) **Ru1085** (10 μM) at 4 °C; (3) **Ru1085** (10 μM) containing 2-Deoxy-D-glucose (50 mM) and oligomycin (5 μM) at 37 °C; (4) **Ru1085** (10 μM) containing NH$_4$Cl (50 mM) at 37 °C; (5) **Ru1085** (10 μM) containing sucrose (50 mM) at 37 °C; and (6) **Ru1085** (10 μM) containing M-βCD (10 mM) at 37 °C for 1 h, respectively. After washed three times by PBS, cells were further incubated solely with **Ru1085** (10 μM) for another 5 h at 37 °C. All cells were then washed with PBS for three times and subjected to NIR-II fluorescence.

**In vitro cytotoxicity studies.** The examined cell lines were seeded in 96-well plates (~5 × 10$^3$ cells/well). Cells were incubated with **Ru1085**, Ru(bpy)$_3$Cl$_2$, cisplatin or 5-ALA at various concentrations for 12 h. The medium was replaced with fresh medium and cells were treated with or without light illumination. For incubation with **Ru1085**, cells were treated with 808 nm laser (0.8 W cm$^{-2}$, 5 min); for incubation with Ru(bpy)$_3$Cl$_2$ or 5-ALA, cells were treated with a 450 nm LED (20 mW cm$^{-2}$, 5 min). After incubation for another 36 h, MTT assay was conducted according to the protocol.

**Lysosomal disruption assay.** A549 cells were seeded on confocal dishes (~5 × 10$^4$ cells/dish). After cells were incubated with **Ru1085** (10 μM) or serum-free medium for 12 h, the medium was replaced with fresh medium and cells were treated with or without 808 nm laser illumination (0.8 W cm$^{-2}$, 5 min). After further incubated for 12 h, cells were stained with AO (5 μM, 20 min), and subsequently visualized by an invert fluorescence microscope ($\lambda_{ex}$ = 488 nm, $\lambda_{em}$ = 520–550 nm for green channel and 590–610 nm for red channel).

**Mitochondrial membrane potential (MMP) assay.** A549 cells were treated as above and stained with JC-1 (10 μg mL$^{-1}$, 20 min), and subsequently visualized by an invert fluorescence microscope ($\lambda_{ex}$ = 488 nm, $\lambda_{em}$ = 520–550 nm for JC-1 monomer; $\lambda_{ex}$ = 530 nm, $\lambda_{em}$ = 590–610 nm for JC-1 aggregate).

**Detection of intracellular ROS generation.** A549 cells were seeded on confocal dishes (~5 × 10$^4$ cells/dish). After cells were incubated with **Ru1085** (10 μM) for 6 h, the medium was replaced by H$_2$-DCFH (20 μM) for further incubation (20 min). Then the medium was replaced with fresh medium, and cells were illuminated with 808 nm laser (0.4 W cm$^{-2}$) for 5 min and subsequently visualized by an invert fluorescence microscope ($\lambda_{ex}$ = 488 nm; $\lambda_{em}$ = 520–550 nm).

**Scavenging intracellular ROS generation.** A549 cells were seeded into 24-well plates (3 × 10$^5$ cells/well). After cells were incubated with **Ru1085** (10 μM) for 4 h, the cells were co-incubated with various ROS scavengers for another 2 h. The ROS scavengers included sodium pyruvate (10 mM), D-mannitol (50 mM), tiron (4,5-Dihydroxy-1,3-benzenedisulfonic acid disodium salt monohydrate, 5 mM), ebselen (2-Phenyl-1,2-benzisoselenazol-3(2H)-one, 50 μM), and sodium azide (50 μM), which were used to scavenge the generated hydrogen peroxide (H$_2$O$_2$), hydroxyl radical (OH$^{\bullet}$), superoxide radical (O$_2^{\bullet-}$), peroxynitrite anion (ONOO$^-$), and $^1$O$_2$, respectively. Then the cells were incubated with H$_2$-DCFH (20 μM) for another 20 min. After that, the medium was replaced with fresh medium, and cells were illuminated with 808 nm laser (0.4 W cm$^{-2}$) for 5 min and subsequently visualized by an invert fluorescence microscope ($\lambda_{ex}$ = 488 nm; $\lambda_{em}$ = 520–550 nm).

**Activation of caspase 3/7 and caspase 1.** A549 cells were treated with serum-free medium (negative control), cisplatin (10 μM), or **Ru1085** (10 μM), respectively. After cells were incubated for 6 h, the medium was replaced with fresh medium, and cells were treated with or without 808 nm laser illumination (0.8 W cm$^{-2}$, 5 min). After further incubated for another 12 h, then cells were treated with caspase 3/7 or caspase 1 activity kit according to the manufacturer's protocol.

**Apoptosis and cell cycle analyses.** For apoptosis tests, A549 cells were incubated with **Ru1085** or serum-free medium for 12 h, and then treated with or without 808 nm laser illumination (0.8 W cm$^{-2}$) for 5 min. After incubation for another 12 h, cells were stained with Annexin V-FITC and PI for 15 min and analyzed with flow cytometry.

For cell cycle analysis, A549 cells were treated as above and then lysed by RNase A (100 μg/mL) for 20 min. After that, cells were stained with PI (0.1 mg mL$^{-1}$) for 15 min and subsequently analyzed cell cycle distribution by flow cytometry.

**Migration and invasion inhibition.** The anti-migration ability was performed by wound-healing assay. A549 cells were seeded into 24-well plates (3 × 10$^5$ cells/well). Cells were incubated with **Ru1085** (10 μM) or serum-free medium for 12 h and then treated with or without laser illumination (808 nm, 0.8 W cm$^{-2}$) for 5 min. Horizontal lines were drawn using micro pipette tips in each well, and subsequently the wound was created. After further incubation for 12 and 24 h, cells were imaged by an invert fluorescence microscope with calcein-AM (5 μM) staining ($\lambda_{ex}$ = 488 nm; $\lambda_{em}$ = 520–550 nm). The wounds area was measured by ImageJ, and wound closure ratio was defined as [1-(wound area at t/original wound area)] × 100%. The anti-invasion capability was tested by Tanswell/Matrigel invasion assay. Transwell inserts were pretreated with Matrigel (200 μg mL$^{-1}$, 100 μL/well). A549 cells were harvested and resuspended in **Ru1085** (10 μM) or serum-free medium, and subsequently added to upper chambers. The upper chambers were placed into the receiver wells, which were supplemented with complete medium. After further incubated for 48 h, the invaded cells were fixed with 4% paraformaldehyde, and washed with PBS and then stained with crystal violet. Transwell inserts visualization was performed on an invert fluorescence microscope. Cell invasion ratios were calculated according to OD$_{590}$.

**Preparation of Ru1085 NPs.** **Ru1085** (1 mg in acetonitrile) and DSPE-PEG5000 (9 mg in secondary water) were stirred overnight. After removing acetonitrile away using nitrogen gas, the mixture was centrifuged (3000 rpm) for 20 min using a 50 kDa centrifugal filter.

**Hemolysis assay.** Fresh blood from mice was obtained in heparinized tubes, centrifuged the mixture at 3000 rpm for 10 min, and followed by adding 20% PBS ($v/v$). **Ru1085** NPs with various concentrations were dissolved in PBS. Water was used as a positive control, while PBS served as a negative control. After incubation

for 1 h, the suspension was centrifuged at 3000 rpm for 5 min. The absorbance at 540 nm was recorded for each sample. The hemolysis assay was calculated using the following formula:

$$\text{Hemolysis Rate}(\%) = \frac{\text{OD}_{sample} - \text{OD}_{PBS}}{\text{OD}_{water} - \text{OD}_{PBS}} \times 100\% \qquad (7)$$

**Animals and tumor model**. All animal experiments involved in this study were conducted in accordance with the Guide for the Care and Use of Laboratory Animals of Central China Normal University. To establish xenografted tumor models, Balb/c nude mice (female, 5 weeks) were subcutaneously injected with A549 cell suspension ($1 \times 10^6$ cells).

**Pharmacokinetics**. The A549 tumor-bearing mice were intravenously injected with **Ru1085** NPs (1 mg Ru/kg). Mice ($n = 3$) were sacrificed at various time points after administration. Blood was collected by eyeball extirpating and kept in heparinized tubes. The amount of Ru in plasma was then measured by ICP-MS.

**In vivo NIR-II fluorescence imaging-guided and monitored antitumor experiments**. After the tumor reached to ~100 mm³, the tumor models were intravenously injected with **Ru1085** NPs (1 mg Ru/kg), and NIR-II fluorescence images (808 nm, 1000 LP, 0.05 W cm$^{-2}$, and 200 ms) were taken using a in vivo NIR-II fluorescence imaging system. For the therapeutic experiments, the tumor models were separated into five groups ($n = 5$), including: (1) **Ru1085** NPs (1 mg Ru/kg) plus laser, (2) **Ru1085** NPs (1 mg Ru/kg), (3) cisplatin (1 mg Pt/kg), (4) PBS plus laser, and (5) PBS. At 24 h post-injection, the tumors (group 1 and 4) were illustrated with 808 nm laser (0.8 W cm$^{-2}$, 10 min) and the temperature was monitored by an IR thermal camera. The NIR-II fluorescence images (808 nm, 1000 LP, 0.05 W cm$^{-2}$, and 200 ms) were taken at various time points till 5 days. The tumor volume and body weight were recorded every 2 days. After 20 days, major organs, tumor and blood were collected to make H&E staining and blood biochemistry analysis.

**Statistical analysis**. Data were presented as means ± standard deviation (s.d.). The sample number ($n$) indicates the number of independent biological samples in each experiment. Statistical differences were compared by unpaired Student's $t$-tests analyzing used GraphPad Prism 6 software.

**Reporting summary**. Further information on research design is available in the Nature Research Reporting Summary linked to this article.

## Data availability

The authors declare that the data supporting the findings of this study are available within the paper and its supplementary information files. Data were available from the corresponding author upon request.

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

## Acknowledgements

This work was supported by National Natural Science Foundation of China (NSFC 22022404, 22074050, and 22125106), Wuhan Scientific and Technological Projects (2019020701011441), the State Key Laboratory of Materials-Oriented Chemical Engineering (KL20-05), the Open Fund of Guangdong Provincial Key Laboratory of Luminescence from Molecular Aggregates, Guangzhou 510640, China (South China University of Technology, 2020-k11ma-06), and the Developmental Fund for Science and Technology of Shenzhen (RCJC20200714114556036).

## Author contributions

Y.S. conceived the project and designed the experiments. Y.X., C.L., Shuai Lu, X.Y., and X.L. designed, synthesized, and characterized the materials. Z.W. carried out the theoretical calculations. Y.X., C.L., and Shuang Liu performed the in vitro and in vivo study. Y.X., X.L., and Y.S. wrote the manuscript. All authors analyzed and discussed the results and have given approval to the final version of the manuscript.

## Competing interests

The authors declare no competing interests.
