## [Peer Review File · Nature Communications]

Construction of Emissive Ruthenium(II) Metallacycle over 1000 nm Wavelength for in vivo Biomedical ApplicationsREVIEWER COMMENTS

Reviewer #1 (Remarks to the Author):

In this paper, Xu et al have presented a novel Ru(II) metallacycle (Ru1085) that emits at a wavelength over 1000 nm, which possesses excellent deep-tissue penetration capability and displays good chemo-phototherapeutic performance. The authors have well characterized the structure of Ru1085, and carried out a series of experiments to demonstrate its properties. Moreover, Ru1085 exhibits much better in vitro anticancer activity against cisplatin-resistant A549 cell line as well as in vivo high-quality NIR-II fluorescence imaging guided chemo-phototherapy against A549 tumor with minimal side effects. This work is original and so far reasonably well executed. The design of long-wavelength emissive metal-based agents will open new opportunities for in vivo biomedical applications and this work will be of interest to many in supramolecular chemistry, fluorescence imaging and biomedicine. In order to recommend the acceptance of this manuscript, the following issues should be addressed:

1. The introduction of NIR-II emissive ligand 1 into the scaffold of Ru1085, the maximum emission wavelength has been red-shifted ~25 nm (from 1060 nm to 1085 nm), why is that? Could the authors give some explanations?
2. Comparing with free ligand 1, Ru1085 has shown much better cellular uptake. Besides the difference in octanol/water partition coefficient, are there other reasons for explaining the more effective cellular uptake for Ru1085?
3. In terms of photostability, negligible changes were observed for Ru1085 after 5 cycles. Did the authors study the photostability after more cycles?
4. Based on the MTT results, Ru1085 presented tremendous antitumor capability with high cytotoxicity values on all selected cancer cells on both normoxic or hypoxic conditions. The phototoxic index is also an important issue for PDT agents. There is some room leaving for the author to optimize the PI value of Ru1085. Whether the author could give some ideas on how to improve the PI value of Ru1085 in the future?
5. The authors explained the reason for good antitumor cytotoxicity under hypoxia as Ru1085 was not primarily dependent $1O_2$. Are there other ROSs under irradiation?
6. After encapsulation of Ru1085 into DSPE-PEG to form the NPs, the measurement of average diameter of Ru1085 by DLS is ~220 nm and by TEM is ~170 nm. Why the value of DLS is larger than that of TEM?
7. The encapsulation rate (e.g., loading efficiency and loading capacity) of Ru1085 NPs should be tested and mentioned.
8. Please describe the sample size of caspase 3/7 and caspase 1 activity tests.
9. The authors proved that the NIR-II fluorescence imaging of the regional vascular system of hindlimb with high resolution using Ru1085 NPs. What about the imaging of the vessels in brain? Could the NPs also achieve a comparable resolution?
10. The detail experimental methods of LA-ICP-MS test and the related instrument should be described in the Methods section.

Reviewer #2 (Remarks to the Author):

Image-guided multimodal diagnosis and therapy are receiving significant attention and have great application potential in preclinical and clinical. In this manuscript, the authors designed a Ru metallacycle (Ru1085) for NIR-II fluorescence imaging guided chemo-phototherapy. And the versatile Ru1085 with emission over 1000 nm (NIR-II) possesses good photothermal conversion efficiency (PTT), ROS generation (PDT) and chemotherapy capability. Overall, this work proposed a novel strategy to develop metallacycle combined with organic fluorophores and the data as presented are solid. The manuscript is recommended for publication after the following issues been addressed.

1. In this manuscript, the Ru1085 was used in vitro experiments, while the Ru1085 NPs was used in vivo. The authors should re-confirm this point since the "Ru1085" and "Ru1085 NPs" are two different agents in terms of octanol/water partition coefficient, hydrophilicity, ROS generation, the PCE, cellular uptake, and so on.

2. The ESR analysis implied the $1O_2$ is one of ROSs (Fig.3d), however, the authors also "speculated that the photocytotoxicity of Ru1085 was independent with $1O_2$ " (line 204)? So which ROS was dependent with the photocytotoxicity of Ru1085?

3. The PPIX works as the photosensitizer in vivo rather than 5-ALA.

4. Toxicity assays should be conducted also with the individual Ru receptor 2 or add the corresponding reference.

5. For table 1, the much better PI value and SI value are both achieve by Ru1085 than that of cisplatin, 5-ALA and Ru(bpy) $_3$ Cl $_2$. The author should add some discussion on how to improve these values in the future work.

6. Whether the author could explain why the SBR for tumor imaging was not as well as for hindlimb vessels imaging?

7. For in vivo tumor imaging or therapy, whether the dose and the power of laser are the same or different?

8. In the section of discussion, the author should point out the potential clinic translation of long-wave emissive metal-based agents in the future.

Response Letter

Reviewer #1:

In this paper, Xu et al have presented a novel Ru(II) metallacycle (**Ru1085**) that emits at a wavelength over 1000 nm, which possesses excellent deep-tissue penetration capability and displays good chemo-phototherapeutic performance. The authors have well characterized the structure of **Ru1085**, and carried out a series of experiments to demonstrate its properties. Moreover, Ru1085 exhibits much better in vitro anticancer activity against cisplatin-resistant A549 cell line as well as in vivo high-quality NIR-II fluorescence imaging guided chemo-phototherapy against A549 tumor with minimal side effects. This work is original and so far reasonably well executed. The design of long-wavelength emissive metal-based agents will open new opportunities for in vivo biomedical applications and this work will be of interest to many in supramolecular chemistry, fluorescence imaging and biomedicine. In order to recommend the acceptance of this manuscript, the following issues should be addressed:

1. The introduction of NIR-II emissive ligand **1** into the scaffold of **Ru1085**, the maximum emission wavelength has been red-shifted ~25 nm (from 1060 nm to 1085 nm), why is that? Could the authors give some explanations?

Response: Thanks for your comments. The spectral red-shift is attributed to the introduction of metal centers, which enhance both intersystem crossing (ISC) and intramolecular charge transfer (ICT) processes to compete with the fluorescence emissions¹.

Reference:

[1] M. L. Saha, X. Yan, P. J. Stang, Photophysical properties of organoplatinum(II) compounds and derived self-assembled metallacycles and metallacages: fluorescence and its applications. *Acc. Chem. Res.* **49**, 2527–2539 (2016).

2. Comparing with free ligand **1**, **Ru1085** has shown much better cellular uptake. Besides the difference in octanol/water partition coefficient, are there other reasons for explaining the more effective cellular uptake for **Ru1085**?

Response: Besides the difference in octanol/water partition coefficient, the following three factors could lead to the enhanced cellular uptake of **Ru1085**. First, **Ru1085** is more lipophilic than free ligand **1** for more easily passage through the cell membrane and being internalized into cells¹. Second, the positively charged **Ru1085** has inherent advantages over the uncharged ligand **1** in entering negative-charged cell membranes². Last, the characterized ring tension of **Ru1085** may also enhance the cellular uptake³. We have added this explanation into the section of discussion.

Reference:

[1] H. H. F. Refsgaard, B. F. Jensen, P. B. Brockhoff, S. B. Padkjær, M. Guldbandt, M. S. Christensen, In silico prediction of membrane permeability from calculated molecular parameters. *J. Med. Chem.* **48**, 805–811 (2005).

[2] H. Duan, S. Nie, Cell-Penetrating Quantum Dots Based on Multivalent and Endosome-Disrupting Surface Coatings. *J. Am. Chem. Soc.* **129**, 3333–3338 (2007).

[3] G. Gasparini, E.-K. Bang, J. Montenegro, S. Matile, Cellular uptake: lessons from

supramolecular organic chemistry. *Chem. Commun.* **51**, 10389-10402 (2015).

3. In terms of photostability, negligible changes were observed for **Ru1085** after 5 cycles. Did the authors study the photostability after more cycles?

Response: Thanks for your good suggestion. We have conducted the photostability test with more cycles. The temperature changes of heating-cooling cycles were monitored with no significant decay even after seven cycle. The following figure has been updated in the revised manuscript.

Figure 3h. Photothermal stability of **Ru1085** (10 μ M) illuminated by 808 nm laser (0.8 W cm⁻²) for seven repeating cycles of heating-cooling.

4. Based on the MTT results, **Ru1085** presented tremendous antitumor capability with high cytotoxicity values on all selected cancer cells on both normoxic or hypoxic conditions. The phototoxic index is also an important issue for PDT agents. There is some room leaving for the author to optimize the PI value of **Ru1085**. Whether the author could give some ideas on how to improve the PI value of **Ru1085** in the future?

Response: Thanks for your constructive suggestion. As you mentioned, there exists some room leaving to optimize the PI value of metallacycle **Ru1085**. In our future project, we aim to improve the PI value based on the strategy of increasing photocytotoxicity and reducing dark cytotoxicity simultaneously. First, it is well-documented that the photocytotoxicity depends on the PDT and PTT effect. Therefore, we could rationally design the ligand moiety through decreasing the singlet-triplet energy gap (ΔE_{ST})¹ and introducing barrier-free rotation² to enhance the ROS generation and photothermal conversion. Second, considering that the dark cytotoxicity was closely related to the Ru(II) acceptor, we could introduce other Ru(II) acceptors with lower cytotoxicity into the metallacycle skeleton³. Third, we could rationally design and self-assemble other non-Ru(II) based metallacycles (such as Ir or Rh) with low dark cytotoxicity. The relevant information has been added in the section of discussion.

Reference:

[1] D. Xi, M. Xiao, J. Cao, L. Zhao, N. Xu, S. Long, J. Fan, K. Shao, W. Sun, X. Yan, X. Peng, NIR light-driving barrier-free group rotation in nanoparticles with an 88.3% photothermal conversion efficiency for photothermal therapy. *Adv. Mater.* **32**, 1907855-1907862 (2020).

[2] S. Xu, Y. Yuan, X. Cai, C. J. Zhang, F. Hu, J. Liang, G. Zhang, D. Zhang, B. Liu, Tuning the singlet-triplet energy gap: a unique approach to efficient photosensitizers with aggregation-induced emission (AIE) characteristics. *Chem. Sci.* **6**, 5824-5830 (2015).

[3] Y. Zhao, L. Zhang, X. Li, Y. Shi, R. Ding, M. Teng, P. Zhang, C. Cao, P. J. Stang, Self-

assembled ruthenium (II) metallacycles and metallacages with imidazole-based ligands and their in vitro anticancer activity. *Proc. Natl. Acad. Sci. USA* **116**, 4090-4098 (2019).

5. The authors explained the reason for good antitumor cytotoxicity under hypoxia as **Ru1085** was not primarily dependent $^1\text{O}_2$. Are there other ROSs under irradiation?

Response: Thanks for your comments. We have conducted ROS scavenging tests to detect cellular ROS types after cells treated with **Ru1085** and laser illumination. As shown in the following figure, hydrogen peroxide (H_2O_2), hydroxyl radical (OH^\bullet), and superoxide radical ($\text{O}_2^{\bullet-}$) were generated in cells. The relevant figure has been added in the revised manuscript.

Supplementary Figure 20. Cellular ROS level in the presence of different ROS quenchers (1: sodium pyruvate, 2: D-mannitol, 3: tiron, 4: ebselen, 5: sodium azide). $\text{H}_2\text{-DCFH}$ was used as ROS indicator.

6. After encapsulation of **Ru1085** into DSPE-PEG to form the NPs, the measurement of average diameter of **Ru1085** by DLS is ~ 220 nm and by TEM is ~ 170 nm. Why the value of DLS is larger than that of TEM?

Response: Thanks for your comments. DLS results were larger than the size obtained from TEM image because of the swelling effect in aqueous solution¹.

Reference:

[1] H. Kakwere, S. Perrier, Orthogonal “Relay” Reactions for Designing Functionalized Soft Nanoparticles. *J. Am. Chem. Soc.* **131**, 1889–1895 (2009).

7. The encapsulation rate (e.g., loading efficiency and loading capacity) of **Ru1085** NPs should be tested and mentioned.

Response: According to your suggestion, we have tested the encapsulation rate of **Ru1085** NPs. The calculated encapsulation efficiency was 31%. We have added the Figure in the revised SI.

Supplementary Figure 26. (a) Absorption spectra of **Ru1085** at various concentration. (b) Plot of the absorbance as a function of the concentration of **Ru1085**. The encapsulation efficiency was calculated to be 28%.

8. Please describe the sample size of caspase 3/7 and caspase 1 activity tests.

Response: Thanks for your comments. The sample size of caspase 3/7 and caspase 1 activity tests was three. We have added the related information in the revised manuscript.

9. The authors proved that the NIR-II fluorescence imaging of the regional vascular system of hindlimb with high resolution using **Ru1085** NPs. What about the imaging of the vessels in brain? Could the NPs also achieve a comparable resolution?

Response: Thanks for your comments. We have conducted the imaging of the vessels in brain using **Ru1085** NPs. As shown in the following figure, the vessels in brain could be visualized from the background skin tissue with signal to background ratio (SBR = 6.9) and ideal spatial resolution (narrow full width at half-maximum (FWHM) = 227 μm). Compared with NIR-II fluorescence imaging of the regional vascular system of hindlimb (SBR = 13.6), the resolution in brain vessels reduced but still considerable, because the skull could limit the quality of fluorescence imaging.

Figure. (a) NIR-II FL images of brain vessels after intravenous injection of **Ru1085** NPs. (b) NIR-II FL intensity profiles (blue line) and Gaussian fit (red line) along the white full line in (a).

10. The detail experimental methods of LA-ICP-MS test and the related instrument should be described in the Methods section.

Response: Thanks for your comments. We have already added the detail experimental methods of LA-ICP-MS test and the related instrument in the updated Methods section. The experiment process was as following: A549 cells were seeded in 24-well culture plates with cell climbing slices for overnight and then incubated with **Ru1085** (10 μM) for 6 h. After washed by PBS w/o Ca/Mg, cells were fixed with 70% cold alcohol solution and washed by water. Then, cells were allowed to adhere on slides to prepare laser ablation inductively coupled plasma mass spectrometry (LA-ICP-MS) test. A laser spot size of 3 μm diameter, 10 $\mu\text{m s}^{-1}$ scan speed, 100 Hz repetition frequency, and laser fluence of $\sim 3 \text{ J cm}^{-2}$ were utilized to perform the test. ICP-MS parameters were as follows: radio frequency power of 1500 W, nebulizer gas flow of 1.25 L min^{-1} , auxiliary gas flow of 1.2 L min^{-1} , plasma gas flow of 15 L min^{-1} . The monitored isotope ^{102}Ru was measured in counting mode. Images integration was performed by the software Igor-based Iolite V3.6. The related instruments were as following: LA-ICP-MS was conducted using

an NWR image laser ablation system (ESI, USA) with a 266 nm Nd: YAG laser and a high performance two volume ablation chamber (TwoVol2) for laser ablation, and a quadrupole ICP-MS (PerkinElmer, NexION 2000D, USA) coupled with the NWR image laser ablation system for element measurement. Above information has been updated in the revised SI.

Reviewer #2:

Image-guided multimodal diagnosis and therapy are receiving significant attention and have great application potential in preclinical and clinical. In this manuscript, the authors designed a Ru metallacycle (**Ru1085**) for NIR-II fluorescence imaging guided chemo-phototherapy. And the versatile **Ru1085** with emission over 1000 nm (NIR-II) possesses good photothermal conversion efficiency (PTT), ROS generation (PDT) and chemotherapy capability. Overall, this work proposed a novel strategy to develop metallacycle combined with organic fluorophores and the data as presented are solid. The manuscript is recommended for publication after the following issues been addressed.

1. In this manuscript, the **Ru1085** was used in vitro experiments, while the **Ru1085** NPs was used in vivo. The authors should re-confirm this point since the “**Ru1085**” and “**Ru1085** NPs” are two different agents in terms of octanol/water partition coefficient, hydrophilicity, ROS generation, the PCE, cellular uptake, and so on.

Response: Thanks for your comments. Following your thoughtful suggestion, we have reconfirmed the point that **Ru1085** was used in vitro experiments, while the **Ru1085** NPs was used in vivo. According to your comments, we have compared the octanol/water partition coefficient ($\log P_{o/w}$), hydrophilicity, $^1\text{O}_2$ quantum yield (Φ_Δ), ROS generation, PCE, and cellular uptake between **Ru1085** and **Ru1085** NPs. After encapsulated in DSPE-PEG5000, $\log P_{o/w}$ reduced from 1.18 to -1.38, and **Ru1085** NPs turned to be hydrophilicity ($\log P_{o/w} < 0$) owing to the hydrophilicity of PEG chains. The $^1\text{O}_2$ quantum yield (Φ_Δ) was calculated to be 1.1 as shown in the following Fig. 1. The photothermal conversion efficiency (PCE) of **Ru1085** NPs was 36.7%, which was comparable with that of **Ru1085** (30.9%). The slight increase of PCE may be attributed to the higher specific heat capacity of water than DMF. The NIR-II fluorescence imaging and ICP-MS analysis results indicated that after encapsulated in DSPE-PEG5000, the cell uptake of **Ru1085** NPs was improved compare to **Ru1085**. The relative data have been updated in the revised manuscript.

Figure 1. (a) $^1\text{O}_2$ generation of **Ru1085** NPs (20 μM) using DPBF as a probe with 808 nm laser illumination (0.8 W cm^{-2}) for various time. (b) Linear calibration curve for the absorbance of DPBF plus **Ru1085** NPs to illumination time.

Figure 2. ROS generation of **Ru1085** (20 μM) illuminated by 808 nm laser (0.8 W cm^{-2}) using $\text{H}_2\text{-DCFH}$ as indicator.

Supplementary Figure 28. Monitored temperature profile (red line) of **Ru1085** NPs illuminated for 300 s and followed by natural cooling, and linear time data versus $-\ln\theta$ (blue line) from the cooling period.

Figure 3. (a) NIR-II FL images of cells incubated with **Ru1085** NPs (10 μM) for 6 h. Scale bar: 20 μm . (b) ICP-MS results of intracellular Ru amount after incubated with **Ru1085** (5 μM) and **Ru1085** NPs (5 μM) for 6 h.

2. The ESR analysis implied the $^1\text{O}_2$ is one of ROSs (Fig.3d), however, the authors also “speculated that the photocytotoxicity of **Ru1085** was independent with $^1\text{O}_2$ ” (line 204)? So which ROS was dependent with the photocytotoxicity of **Ru1085**?

Response: Thanks for your comments. We have conducted ROS scavenging tests to detect cellular ROS types after cells treated with **Ru1085** and laser illumination. As shown in the following figure, hydroxyl radical (OH^\cdot) was produced most under laser illumination. Besides, hydrogen peroxide (H_2O_2) and superoxide radical ($\text{O}_2^{\cdot-}$) were also generated in cells. As we speculated, little $^1\text{O}_2$ was produced to cause phototoxicity effect. The relevant figure has been added in the revised manuscript.

Supplementary Figure 20. Cellular ROS level in the presence of different ROS quenchers (1: sodium pyruvate, 2: D-mannitol, 3: tiron, 4: ebselen, 5: sodium azide). $\text{H}_2\text{-DCFH}$ was used as ROS indicator.

3. The PPIX works as the photosensitizer in vivo rather than 5-ALA.

Response: Thanks for your suggestion. We are totally agreed with PPIX was more suitable for in vivo applications. In our current work, we refer previous work using 5-ALA as the photosensitizer^{1,2}. In our future work, we will use PPIX as the photosensitizer for comparison.

Reference:

1. Zhou, Z. Liu, J. Huang, J. Rees, T. W. Wang, Y. Wang, H. Li, X. Chao, H. & Stang, P. J. A self-assembled Ru–Pt metallacage as a lysosome-targeting photosensitizer for 2-photon photodynamic therapy. *Proc. Natl. Acad. Sci. USA* **116**, 20296-20302 (2019)
2. Ung, P. Clerc, M. Huang, H. Qiu, K. Chao, H. Seitz, M. Boyd, B. Graham, B. & Gasser, G. Extending the excitation wavelength of potential photosensitizers via appendage of a kinetically stable terbium(III) macrocyclic complex for applications in photodynamic therapy. *Inorg. Chem.* **56**, 7960–7974 (2017)

4. Toxicity assays should be conducted also with the individual Ru receptor 2 or add the corresponding reference.

Response: Thanks for your comments. As the toxicity of the individual Ru(II) receptor **2** has been reported, we have added the corresponding reference (44. Zhao, Y. Zhang, L. Li, X. Shi, Y. Ding, R. Teng, M. Zhang, P. Cao, C. & Stang, P. J. Self-assembled ruthenium (II) metallacycles and metallacages with imidazole-based ligands and their in vitro anticancer activity. *Proc. Natl. Acad. Sci. USA* **116**, 4090–4098 (2019).) in the revised manuscript.

5. For table 1, the much better PI value and SI value are both achieved by **Ru1085** than that of cisplatin, 5-ALA and Ru(bpy)₃Cl₂. The author should add some discussion on how to improve these values in the future work.

Response: Thanks for your good suggestion. As our response 4 to Reviewer 1, there exists some room leaving to optimize phototoxic index (PI) value and selectivity index (SI) value. In our future project, we aim to improve the PI value based on the strategy of increasing photocytotoxicity and reducing dark cytotoxicity simultaneously. First, it is well-documented that the photocytotoxicity depends on the PDT and PTT effect. Therefore, we could rationally design the ligand moiety through decreasing the singlet-triplet energy gap (ΔE_{ST})¹ and introducing barrier-free rotation² to enhance the ROS generation and photothermal conversion. Second, considering that the dark cytotoxicity was closely related to the Ru(II) acceptor, we could introduce other Ru(II) acceptors with lower cytotoxicity into the metallacycle skeleton³. For improving SI value, special peptides⁴ or ligand could be introduced into the skeleton to improve the specific uptake of tumor cells. The relevant information has been added in the section of discussion.

Reference:

[1] D. Xi, M. Xiao, J. Cao, L. Zhao, N. Xu, S. Long, J. Fan, K. Shao, W. Sun, X. Yan, X. Peng, NIR light-driving barrier-free group rotation in nanoparticles with an 88.3% photothermal conversion efficiency for photothermal therapy. *Adv. Mater.* **32**, 1907855-1907862 (2020).

[2] S. Xu, Y. Yuan, X. Cai, C. J. Zhang, F. Hu, J. Liang, G. Zhang, D. Zhang, B. Liu, Tuning the singlet-triplet energy gap: a unique approach to efficient photosensitizers with aggregation-induced emission (AIE) characteristics. *Chem. Sci.* **6**, 5824-5830 (2015).

[3] Y. Zhao, L. Zhang, X. Li, Y. Shi, R. Ding, M. Teng, P. Zhang, C. Cao, P. J. Stang, Self-assembled ruthenium (II) metallacycles and metallacages with imidazole-based ligands and their in vitro anticancer activity. *Proc. Natl. Acad. Sci. USA* **116**, 4090–4098 (2019).

[4] C. Chen, J. Hu, P. Zeng, Y. Chen, H. Xu, J. R. Lu, High cell selectivity and low-level antibacterial resistance of designed amphiphilic peptide G(IKK)3I-NH₂. *ACS Appl. Mater. Interfaces*, **6**, 16529–16536 (2014).

6. Whether the author could explain why the SBR for tumor imaging was not as well as for hindlimb vessels imaging?

Response: Thanks for your comments. The vessels imaging was conducted immediately after intravenous-injection, while the tumor imaging (at the highest SBR time-point) was performed at 24 h post-injection. Owing to the rapid process of hindlimb vessels imaging, **Ru1085** NPs did not accumulate in the surrounding normal tissues, so that the SBR was high; while in tumor imaging, **Ru1085** NPs accumulated in both the tumor site and surrounding normal tissues. The different accumulation in tumor site and normal tissues reflects SBR. Therefore, the SBR of tumor imaging was higher than that of hindlimb vessels imaging.

7. For in vivo tumor imaging or therapy, whether the dose and the power of laser are the same or different?

Response: Thanks for your comments. The dose and power of laser were different in tumor imaging or therapy. The power density used in tumor imaging was 0.05 W cm⁻². The power density used in tumor imaging was 0.8 W cm⁻². The related information has been added in the

revised SI.

8. In the section of discussion, the author should point out the potential clinic translation of long-wave emissive metal-based agents in the future.

Response: Thanks for your good suggestion. We have added the potential clinic translation in the revised manuscript: “Therefore, the construction of long-wavelength emissive metal-based agents could enhance optical penetration for improving phototherapeutic efficacy in deep and/or solid tumors. Besides, taking the intrinsic advantage of long-wavelength emission, they could be utilized as a universal platform for visualizing the delivery, targeting, pharmacokinetics and distribution of metal-based agents through fluorescence imaging. Finally, such a system could provide real-time feedback to the treatment, and facilitate the clinic translation of metal agents in synergistic chemo-phototherapy in the future.”

REVIEWERS' COMMENTS

Reviewer #1 (Remarks to the Author):

The authors have well addressed the comments raised by me and the other referee. The revised manuscript is substantially improved over previous one. The manuscript is recommended for the publications in Nat. Commun. One tiny concern from me is about the title, which the authors called "Self-assembly of Emissive....". To the best of my knowledge, "Self-assembly" means package of compound itself in order, normally without the use of other cargoes or assistants. In this paper, the Ru1085 NPs was formed by nanoencapsulation of Ru1085 within DSPE-PEG. This may be not self-assembly, but more like Nanoaggregation or "Nanoprecipitation".

Reviewer #2 (Remarks to the Author):

The revised manuscript has significantly improved and my concerns have been completely addressed by the authors. Therefore, I recommend its publication as is.

REVIEWERS' COMMENTS

Reviewer #1:

The authors have well addressed the comments raised by me and the other referee. The revised manuscript is substantially improved over previous one. The manuscript is recommended for the publications in Nat. Commun. One tiny concern from me is about the title, which the authors called "Self-assembly of Emissive....". To the best of my knowledge, "Self-assembly" means package of compound itself in order, normally without the use of other cargoes or assistants. In this paper, the Ru1085 NPs was formed by nanoencapsulation of Ru1085 within DSPE-PEG. This may be not self-assembly, but more like Nanoaggregation or "Nanoprecipitation".

Response: Thanks for your constructive suggestion. We have revised the title and related description from "self-assembly" into "construction" in the revised manuscript and we have highlighted the revision part.

Reviewer #2:

The revised manuscript has significantly improved and my concerns have been completely addressed by the authors. Therefore, I recommend its publication as is.

Response: Thanks for your comments.